# Rethinking Entropy in Test-Time Adaptation: The Missing Piece from Energy Duality

**Mincheol Park**[1]    **Heeji Won**[2]    **Won Woo Ro**[3]*    **Suhyun Kim**[4]*

[1]Samsung Advanced Institute of Technology,    [1]Samsung Electronics,
[2]Korea University,    [3]Yonsei University,    [4]Kyung Hee University
danimc.park@samsung.com, gmlwl1026@korea.ac.kr,
wro@yonsei.ac.kr, dr.suhyun_kim@gmail.com

## Abstract

Test-time adaptation (TTA) aims to preserve model performance under distribution shifts. Yet, most existing methods rely on entropy minimization for confident predictions. This paper re-examines the sufficiency of entropy minimization by analyzing its dual relationship with energy. We view energy as a proxy for likelihood, where lower energy indicates higher observability under the learned distribution. We uncover that entropy and energy are tightly associated, controlled by the model's confidence or ambiguity, and show that simultaneous reduction of both is essential. Importantly, we reveal that entropy minimization alone neither ensures energy reduction nor supports reliable likelihood estimation, and it requires explicit discriminative guidance to reach zero entropy. To combat these problems, we propose a twofold solution. First, we introduce a likelihood-based objective grounded in energy-based models, which reshape the energy landscape to favor test samples. For stable and scalable training, we adopt sliced score matching—a sampling-free, Hessian-insensitive approximation of Fisher divergence. Second, we enhance entropy minimization with a cross-entropy that treats the predicted class as a target to promote discriminability. By counterbalancing entropy and energy through the solution of multi-objective optimization, our unified TTA, ReTTA, outperforms existing entropy- or energy-based approaches across diverse distribution shifts.

## 1 Introduction

Deep learning models are increasingly ubiquitous in cutting-edge technologies such as autonomous vehicles [21, 17] and biomedical science [1, 22]. A key assumption behind their success is that test data comes from the same distribution as training data. However, this assumption is often broken in practice. Test data can be subject to degeneration, referred to as covariate shifts, such as changes in lighting due to weather conditions or unexpected noise caused by sensor degradation [14]. Unfortunately, the decision-making of models degrades due to the distribution shifts [13]. This poses a substantial challenge for the practical deployment of pre-trained models.

Test-time adaptation (TTA) has been proposed to yield more reliable decision-making under distribution shifts. An early TTA method centered on minimizing the entropy of test data [34], assuming models could separate classes well [11]. This key idea became the cornerstone of state-of-the-art methods [9, 20, 26, 27, 40], fueling the evolution of TTA. These modern approaches adopt proxy techniques in parallel, such as data filtering [26] or pseudo-labeling [10, 23, 35], because data streams experience dynamic shifts [38] and high correlation (non-i.i.d.) [2] in practice. Still, the core principle of minimizing entropy remains unchanged, even as many efforts are made to address the limitations of optimizing entropy alone.

---

*Co-corresponding author.

39th Conference on Neural Information Processing Systems (NeurIPS 2025).

In this study, we raise a crucial question: *Is minimizing entropy truly sufficient as the core objective for TTA?* Our revisit to entropy through its dual, energy [18], which reflects the likelihood of being in-distribution, reveals a key gap: the lack of momentum to minimize energy for better likelihood estimation in the test domain. More recently, an energy-based TTA method [39] aims to reduce energy overall, but it overlooks the advantages of minimizing entropy, which enhances discriminability.

This paper argues that the complete reduction of entropy and energy is crucial. However, even for data with low confidence, achieving zero entropy is difficult, even when entropy and energy are minimized simultaneously. Our test supports this by analyzing the distribution of entropy and energy in test data (Figure 1). The distribution reflects whether the logit would be a confident or ambiguous prediction, and implies discrete bands following a log-shaped curve (Figure 1(d)). Here, the lower energy and entropy correlate with correct predictions. However, this observation also reveals that, with only the two objectives, test data cannot transition between these bands to improve accuracy. This, in short, signifies the need for an additional objective to guide the model toward discriminability.

This paper introduces *ReTTA*, a novel unified TTA based on both entropy and energy with two objectives. The first objective is to utilize energy-based models (EBMs) [18], where the marginal density of data is modeled by an energy function, typically defined as the LogSumExp of the model's logit. EBMs allow the model to reshape its likelihood in response to data [24], with training focusing on maximizing the density by lowering the energy of test data and raising the energy of generated samples [39]. However, sampling during EBM training can be unstable [5]. We approximate this process using a first-order method. This reformulation allows for minimizing Fisher divergence between the test and model distributions by aligning their *scores* [31]. This sampling-free objective is well-suited for TTA. Precisely, to avoid unstable loss due to abnormal Hessian values, we adopt *sliced score matching* (SSM), which provides a scalar comparative loss for TTA [33].

The second objective is to achieve complete entropy minimization (EM) by incorporating a discriminative objective that guides the model's prediction toward a single class. To this end, we use cross-entropy, targeting the most probable class, which we define as the *targeted class convergence* (TCC) loss. By combining EM loss with SSM and TCC, we form the unified ReTTA loss. Here, entropy and energy optimization should be handled carefully because the lack of supervision can hinder convergence, especially given the unpredictable nature of distribution shifts. To address the challenge of balancing entropy and energy optimization, we propose a self-adjusting coefficient, where energy optimization is adaptively adjusted relative to EM, regardless of the type of distribution shift. Extensive experimental evaluations demonstrate that ReTTA outperforms TTA approaches that rely solely on entropy or energy optimization.

Our contributions are summarized as follows:

- We confirm that minimizing entropy alone is insufficient for estimating the test distribution's likelihood, emphasizing the necessity for simultaneous entropy and energy minimization in TTA. We address this with a sampling-free energy adaptation loss (SSM), which, combined with EM, directly maximizes likelihood.

- We establish that successful TTA requires energy reduction and convergence to the lowest entropy. We propose the targeted class convergence (TCC) loss, using cross-entropy, and integrate it with EM and SSM in a novel unified EM objective, ReTTA.

- We propose a self-adjusting coefficient to counterbalance the optimization of entropy and energy, effectively addressing challenges such as unpredictable distribution shifts. Evaluations demonstrate that ReTTA adaptively works on various corruption data and performs well.

## 2 Related Work

**Test-time adaptation (TTA).**   TTA aims to improve model generalizability under distribution shifts. This is achieved by updating model parameters using test data, without access to the training dataset. Numerous TTA techniques have been proposed, including pseudo-labeling [23, 10, 35], calibration of normalization layers [8, 28, 38, 27], consistency-based regularization [30], prototype alignment [15], low-rank mixtures of experts [19], and energy adaptation [39]. Among these, EM [34, 26, 27, 20, 9] remains a widely adopted objective, encouraging confident predictions during adaptation. However, Boudiaf et al. [2] highlight the failure of EM when the test stream lacks diversity. More recently, Choi et al. [3] point out the limitations of relying solely on EM in such unpredictable scenarios,

emphasizing the importance of energy. Building upon these insights, we offer a new perspective by showing why EM, while useful, is insufficient on its own. Additionally, we explore the energy-entropy relationship and argue that, to improve the effectiveness of entropy minimization, it is essential to reduce energy and boost the model's discriminability concurrently.

**Energy-based models (EBMs).** EBMs [18] are a class of non-normalized probabilistic models with an intractable normalizing constant. They use stochastic approximations to estimate this constant, offering flexibility in parameterization and enabling the modeling of a wide range of distributions [12]. Through these approximations, EBMs generate data via energy functions, without relying on an explicit neural network. This flexibility has led to applications in tasks such as image generation [4, 6], domain adaptation [41, 37], and domain generalization [7, 36]. Recently, energy-based methods for TTA have focused on reducing energy within the model's distribution to enhance generalizability [38]. However, the method ignores the direction of energy alignment and requires multiple sampling iterations due to the intractable constant. While adaptive energy adaptation [3] attempts to address these issues, it relies on heuristics and mini-batch configurations. In contrast, our work introduces a more scalable approach through Sliced Score Matching (SSM) [33], providing a sampling-free objective that improves TTA while avoiding the problems [5] in training EBMs.

# 3 Analytical Motivation and Observation in Entropy Minimization

**Preliminaries.** Let the source dataset $\mathcal{D}_s$ be sampled from the training data distribution $p_s(\mathbf{x}, y)$, and the target dataset $\mathcal{D}_t$ from the test data distribution $p_t(\mathbf{x}, y)$. A discriminative model $f_\theta : \mathbb{R}^D \mapsto \mathbb{R}^K$, parameterized by $\theta$, which maps data $\mathbf{x} \in \mathbb{R}^D$ to $K$ real-valued outputs, is trained by maximizing the log-posterior $\log p(\theta|\mathcal{D}_s)$ for the source dataset. During testing, the model $f_\theta$ infers the label $y_t$ for unseen test (target) data $\mathbf{x}_t$ from the $K$ classes by marginalizing over the parameters $\theta$ as:

$$p(y_t|\mathbf{x}_t; \mathcal{D}_s) = \int p(y_t|\mathbf{x}_t; \theta)p(\theta|\mathcal{D}_s)d\theta, \tag{1}$$

where $(\mathbf{x}_t, y_t) \in \mathcal{D}_t$. Covariate shift occurs due to the shift in the marginal distribution of data, i.e., $p_s(\mathbf{x}) \neq p_t(\mathbf{x})$. When covariate shifts exist, the joint distribution also differs, i.e., $p_s(\mathbf{x}, y) \neq p_t(\mathbf{x}, y)$, which compromises inference in Eq. 1 by causing a mismatch in likelihoods and leads to degraded accuracy. As a workaround, many TTA methods [20, 26, 27, 34] attempt to update the parameters $\theta$ applying EM. In Section 3.1 and 3.2, we discuss what is missing in the EM during TTA.

## 3.1 Rethinking Entropy Minimization

The mitigation of the covariate shift is key to the success of TTA. Specifically, given the model parameterized by $\theta$, which estimates the source distribution $p_s(\mathbf{x}, y)$, one promising approach is to update $\theta$ to maximize the likelihood of $p_t(\mathbf{x})$ when test data is observed, i.e., $p_s(\mathbf{x}) \simeq p_t(\mathbf{x})$.

In this regard, we account for $p(\mathbf{x})$ as the sum of its factorized component $y$, i.e., $p(\mathbf{x}) = \sum_y p(\mathbf{x}, y)$. However, at test time, the label $y$ is unknown. TTA approaches treat the probable classes as potential labels, assuming that the model, having once maximized the log-likelihood on the source domain, has strong discriminative power [11]. Therefore, the probable classes are determined by the model's output probabilities $\mathbf{p}$, which is computed by applying the Softmax to the logits:

$$\mathbf{p}(\mathbf{x}) = \big[p(y = 1|\mathbf{x}; \theta), \dots, p(y = K|\mathbf{x}; \theta)\big]; \quad p(y|\mathbf{x}; \theta) = \frac{\exp(f_\theta(\mathbf{x})[y])}{\sum_k \exp(f_\theta(\mathbf{x})[k])}, \tag{2}$$

where $\mathbf{p}(\mathbf{x}) \in \mathbb{R}^K$ and $f_\theta(\mathbf{x})$ represents the logit of the data $\mathbf{x}$. Here, we note whether the likelihood of the data, marginalized over these probable classes, is maximized by minimizing the entropy of the Softmax $\mathbf{p}$. To this end, it is necessary to introduce a quantity that quantifies how the data likely belongs to the marginal distribution parameterized by $\theta$. We use the energy [18, 39], $E_\theta(\mathbf{x})$, which maps data or its logit to a deterministic scalar by summing over the probable classes, as defined by:

$$E_\theta(\mathbf{x}) := -\log \sum_k \exp(f_\theta(\mathbf{x})[k]). \tag{3}$$

This log partition function, also defined as $E_\theta(\mathbf{z}) = -\texttt{LogSumExp}(\mathbf{z})$ with $\mathbf{z} = f_\theta(\mathbf{x})$, indicates that a larger negative value represents more likely (or highly observable) data under the distribution $p_\theta(\mathbf{x})$.

Now, we focus on the relationship between energy and entropy. We note that they form a conjugate pair, exhibiting a *duality* that helps us understand the trajectory of energy during minimizing entropy.

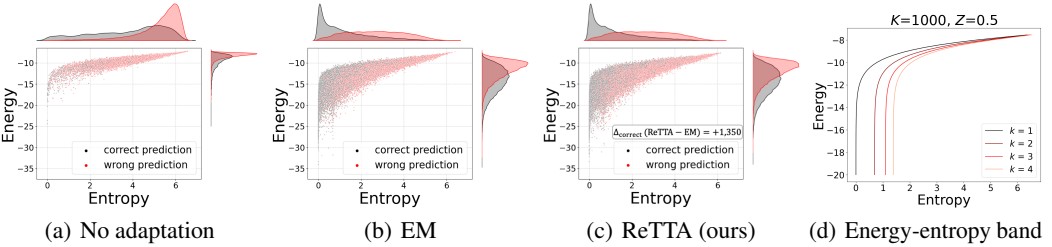

| (a) No adaptation | (b) EM | (c) ReTTA (ours) | (d) Energy-entropy band |

Figure 1: Distribution of test data based on energy and entropy values, and visualization of energy-entropy bands. **(a)** No adaptation: inference without TTA. **(b)** EM: results from the state-of-the-art EM method [27]. **(c)** ReTTA (ours): unified TTA integrating EM with the two objectives from Section 4 (1,350 more correct samples than EM). To observe this phenomenon, ResNet-50 (BN) is applied to the contrast corruption (severity 5) from ImageNet-C [14], following [27]. **(d)** Energy–entropy band: energy-entropy curves with respect to the number of classes $K$ and the secondary logit $Z$.

**Lemma 1** (Conjugate Relation). *Suppose* $\mathbf{z}$ *represents the model's logit, and* $\mathbf{g}$ *denotes the gradient of the concave function* $E_\theta$ *with respect to the logit* $\mathbf{z}$. *The concave conjugate of* $E_\theta(\mathbf{z})$ *is defined as* $E_\theta^*(\mathbf{g}) = \min_{\mathbf{z}}\{\mathbf{g}^T\mathbf{z} - E_\theta(\mathbf{z})\}$. *Then, the gradient* $\mathbf{g}$ *corresponds negatively to the Softmax, i.e.,* $\mathbf{g} = \nabla_{\mathbf{z}}E_\theta(\mathbf{z}) = -\mathbf{p}(\mathbf{x})$, *and the conjugate function* $E_\theta^*(\mathbf{g})$ *becomes the negative entropy of* $\mathbf{p}(\mathbf{x})$:

$$E_\theta^*(\mathbf{g}) = H(\mathbf{p}) = -\mathbf{p}(\mathbf{x})^T \log \mathbf{p}(\mathbf{x}). \tag{4}$$

Eq. 4 holds in reverse when considering the conjugate of negative entropy (for clarity, we use "entropy" to refer to negative entropy). Thus, both functions exhibit bi-duality, with each being the conjugate of the other. Building on the bi-duality, we consider the following Fenchel duality.

**Lemma 2** (Fenchel-Moreau Theorem). *Primal function* $E_\theta(\mathbf{z})$ *and its conjugate function* $E_\theta^*(\mathbf{g})$ *exhibit bi-duality. The primal function can be completely recovered from its conjugate function* $E_\theta^*(\mathbf{g})$ *as* $E_\theta(\mathbf{z}) = \min_{\mathbf{g}}\{\mathbf{g}^T\mathbf{z} - E_\theta^*(\mathbf{g})\}$. *Therefore, energy and entropy satisfy the following relationship:*

$$E_\theta(\mathbf{z}) = \min_{\mathbf{p}}\{-\mathbf{p}^T\mathbf{z} - H(\mathbf{p})\}. \tag{5}$$

**Analytical motivation.** The duality in Eq. 5 provides valuable insight. When entropy is minimized, $H(\mathbf{p}) \to 0$, the model's output $\mathbf{p}$ becomes a one-hot vector. Accordingly, the product $\mathbf{p}^T\mathbf{z}$ approaches the logit of the most confident class $k^*$, i.e., $E_\theta(\mathbf{z}) = -z_{k*}$, which corresponds to the overall energy. In other words, minimizing entropy does not provide a clear momentum to reduce the overall energy. In short, *EM updates model parameters to increase confidence for the confident classes in test data, but lacks an objective to maximize the likelihood of the marginal distribution.*

### 3.2 Observation for Energy and Entropy Relationship

In this section, we examine another problem of minimizing entropy alone. We confirm this by visually observing how the energy and entropy of test data change under the influence of EM. Figure 1(a) shows that, when TTA is not applied, test data exhibit high entropy and energy, with the distribution concentrated around these high values. When EM works (Figure 1(b)), test data converge toward a region where entropy approaches zero, with energy moving toward larger negative values.

Intriguingly, the distribution of test data with respect to energy and entropy suggests a *log-shaped* relation, with their outermost curve acting as an upper bound. Here, we view this curve as a new perspective for understanding the tight interaction between energy and entropy. Specifically, as we found, by applying restricting conditions to the model's logits, we can define a function capable of interpreting the distribution of energy and entropy.

**Theorem 1.** *Suppose the logit of the model* $f_\theta$ *is defined over* $K$ *classes, where* $k$ *classes are assigned a primary logit* $z^*$, *with strong influence, and the remaining* $K - k$ *classes share a singular logit* $Z$ *with minimal influence. Then, the closed-form equation for the energy-entropy relationship based on the conditioned logits is given by:*

$$H(E_\theta) = -(1 - C(k)e^{E_\theta}) \log\left(\frac{1 - C(k)e^{E_\theta}}{k}\right) - C(k)e^{E_\theta}(Z + E_\theta), \tag{6}$$

where $E_\theta \in \mathbb{R}^-$ denotes the energy, and $H \in [0, \log K]$ represents the entropy. $C(k)$ is a variable defined by $C(k) = (K - k)e^Z$.

From Figure 1(d), Eq. 6 shows that the energy-entropy relationship forms "bands," which represent sets of closely spaced function values that the conditioned logits can occupy, depending on the discrete value of the remaining classes $k$. These bands thus make it easy to infer the possible values of the logit for the test data. For instance, Figure 1(b) shows that if multiple primary logits exist (i.e., $k = 2, 3$) with distinct values, the data can be distributed across the bands, or when the logits are singular, the data will lie on each band. Here, we note one phenomenon: *test data near the zero-entropy and low-energy band ($k = 1$) appear to be corrected.*

**Motivation.** Our motivation stems from the fact that minimizing entropy alone makes it difficult for the Softmax of the logits to converge to the zero-entropy region of the band, $k = 1$. This difficulty arises from the non-zero probabilities in the Softmax, which cause different convergences for the logits of each class [3]. This issue becomes particularly apparent in TTA, where the model observes and updates the data once during inference [34]. Therefore, *an additional goal should be to guide the logits toward the $k = 1$ band explicitly, where entropy approaches zero.*

## 4 Methodology

Building on both motivations, we present two TTA objectives in conjunction with EM: (1) maximizing the likelihood of marginal distribution, and (2) guiding the logits of $\mathbf{x}_t$ toward a zero-entropy region.

**Energy-based models.** We model the marginal distribution using EBMs [18]. Specifically, we define the joint distribution over test data $\mathbf{x}_t$ and a possible class $y_t$ based on the model's logit as $p_\theta(\mathbf{x}_t, y_t) = \exp(f_\theta(\mathbf{x}_t)[y_t])/Z(\theta)$, where $Z(\theta)$ denotes the normalizing constant [12]. Marginalizing out the class variable $y_t$, we obtain the marginal distribution over $\mathbf{x}_t$:

$$p_\theta(\mathbf{x}_t) := \frac{\exp(-E_\theta(\mathbf{x}_t))}{Z(\theta)}, \tag{7}$$

where the energy $E_\theta(\mathbf{x}_t)$ follows Eq. 3. The normalizing constant $Z(\theta) = \int_{\mathbf{x}_t} \exp(-E_\theta(\mathbf{x}_t))d\mathbf{x}_t$ is intractable, which poses a challenge in optimizing the log-likelihood of $p_\theta(\mathbf{x}_t)$: $\max_\theta \mathbb{E}_{p_t}[\log p_\theta(\mathbf{x}_t)]$. We revisit this challenge in Section 4.1 and introduce a proxy objective for stable parameter updates.

### 4.1 Sliced Score Matching Loss

The derivative of the expected log-density $\mathbb{E}_{p_t}[\log p_\theta(\mathbf{x}_t)]$ encourages the model to decrease the energy of the test data while increasing the energy of confabulations (samples generated by the model). Formally, the derivative is given by:

$$\nabla_\theta \mathbb{E}_{p_t}[\log p_\theta(\mathbf{x}_t)] = \mathbb{E}_{p_\theta}[\nabla_\theta E_\theta(\mathbf{x}_t)] - \mathbb{E}_{p_t}[\nabla_\theta E_\theta(\mathbf{x}_t)]. \tag{8}$$

The first expectation term $\mathbb{E}_{p_\theta}[\cdot]$ in Eq. 8 involves generating samples (confabulations) from the model distribution $p_\theta(\mathbf{x}_t)$, typically achieved through a Markov Chain Monte Carlo (MCMC) method, e.g., Gibbs sampling. Among various MCMC techniques, Langevin dynamics is widely adopted as a representative gradient-based sampling approach [5, 16, 39]:

$$\mathbf{x}_t^{i+1} = \mathbf{x}_t^i - \frac{\mu^2}{2}\nabla_{\mathbf{x}_t} E_\theta(\mathbf{x}_t^i) + \mu\boldsymbol{\epsilon}, \quad \boldsymbol{\epsilon} \sim \mathcal{N}(\mathbf{0}, \mathbf{I}_D), \tag{9}$$

where the Markov chain is initialized from the test data, i.e., $\mathbf{x}_t^0 = \mathbf{x}_t$. Here, $\mu$ controls the step size, and $\boldsymbol{\epsilon}$ is Gaussian noise added at each iteration. However, Eq. 9 alone is often insufficient for stable optimization of Eq. 8 [5], and it typically requires repeated iterations. To combat the instability and inefficiency associated with MCMC-based sampling, we adopt an alternative [32]. We leverage the fact that a single-step Langevin update, applied to data sampled from the true distribution $p_t(\mathbf{x}_t)$, provides a good approximation to the gradient of the Fisher divergence between $p_t(\mathbf{x}_t)$ and $p_\theta(\mathbf{x}_t)$.

**Lemma 3.** *A one-step Langevin update initialized from $\mathbf{x}_t \sim p_t(\mathbf{x}_t)$ approximates the gradient of the Fisher divergence between the true distribution $p_t(\mathbf{x}_t)$ and the model distribution $p_\theta(\mathbf{x}_t)$ parameterized by $\theta$, as follows:*

$$\nabla_\theta \mathbb{E}_{p_t}[\log p_\theta(\mathbf{x}_t)] \simeq \frac{\mu^2}{2}\nabla_\theta D_F(p_t(\mathbf{x}_t)||p_\theta(\mathbf{x}_t)) + o(\mu^2), \tag{10}$$

*where $D_F(p||q)$ is the Fisher divergence, and $o(\mu^2)$ denotes higher-order term with respect to $\mu^2$.*

This formulation justifies the use of the Fisher divergence as a surrogate objective for likelihood-based training in EBMs when employing a one-step Langevin update. In particular, the Fisher divergence is also known as *score matching*, and is defined as:

$$D_F(p_t(\mathbf{x}_t)||p_\theta(\mathbf{x}_t)) = \mathbb{E}_{p_t}[||\nabla_{\mathbf{x}_t} \log p_t(\mathbf{x}_t) - \nabla_{\mathbf{x}_t} \log p_\theta(\mathbf{x}_t)||^2] \tag{11}$$

$$\simeq \mathbb{E}_{p_t}[\text{Tr}(\nabla^2_{\mathbf{x}_t} \log p_\theta(\mathbf{x}_t)) + \frac{1}{2}||\nabla_{\mathbf{x}_t} \log p_\theta(\mathbf{x}_t)||^2], \tag{12}$$

where $\nabla_{\mathbf{x}} \log p(\mathbf{x})$ is the score function, which characterizes how the log-density of $p(\mathbf{x})$ varies with respect to $\mathbf{x}$. Unlike Eq. 8, this score matching in Eq. 11 enables a sampling-free optimization via a simple Monte Carlo estimator based solely on empirical averages over the test data. However, computing score matching requires evaluating the trace of the Hessian $\nabla^2_{\mathbf{x}_t}$ of the model's log-density in Eq. 12. This term is costly to compute [25] and can be overly sensitive to the sharp local curvature. Therefore, the use of score matching may become limited in high-dimensional data.

In this paper, we leverage Sliced Score Matching (SSM) [33], a variant of score matching that scales well to high-dimensional data. The key idea is to match inner products of score functions along randomly sampled directions, instead of matching the full score values directly:

$$D_{SF}(p_t(\mathbf{x}_t)||p_\theta(\mathbf{x}_t)) = \mathbb{E}_{p_t,p(\mathbf{v})}[||\mathbf{v}^T \nabla_{\mathbf{x}_t} \log p_t(\mathbf{x}_t) - \mathbf{v}^T \nabla_{\mathbf{x}_t} \log p_\theta(\mathbf{x}_t)||^2], \tag{13}$$

$$= \mathbb{E}_{p_t,p(\mathbf{v})}[\mathbf{v}^T \nabla^2_{\mathbf{x}_t} \log p_\theta(\mathbf{x}_t)\mathbf{v} + \frac{1}{2}||\mathbf{v}^T \nabla_{\mathbf{x}_t} \log p_\theta(\mathbf{x}_t)||^2], \tag{14}$$

where $D_{SF}(p||q)$ is the sliced Fisher divergence and $p(\mathbf{v})$ is chosen as a normal distribution $\mathcal{N}(\mathbf{0}, \mathbf{I}_D)$. Other valid choices for $p(\mathbf{v})$ should satisfy $\mathbb{E}_{p(\mathbf{v})}[\mathbf{v}\mathbf{v}^\top] > 0$ and $\mathbb{E}_{p(\mathbf{v})}[\|\mathbf{v}\|_2^2] < \infty$ [33]. Building on this formulation, we construct the following unbiased estimator as a proxy for maximizing the log-density of $p_\theta(\mathbf{x}_t)$, defined by Eq. 7:

$$\ell_{SSM}(\theta) = \frac{1}{|\mathcal{B}_t|} \sum_{\mathbf{x}_t \in \mathcal{B}_t} \left[ \sum_{i=1}^{D} \sum_{j=1}^{D} \frac{\partial^2 E_\theta(\mathbf{x}_t)}{\partial x_t^i \partial x_t^j} v^i v^j + \frac{1}{2} \sum_{i=1}^{D} \left( \frac{\partial E_\theta(\mathbf{x}_t)}{\partial x_t^i} v^i \right)^2 \right], \tag{15}$$

where $\mathcal{B}_t$ represents a mini-batch of test data $\mathbf{x}_t$, sampled by $\mathcal{D}_t$. Eq. 15 defines our first objective that concentrates on enhancing the TTA of EM.

### 4.2 Targeted Class Convergence Loss

One key challenge remains: when Softmax in Eq. 2 is applied to the logits with respect to the data $\mathbf{x}_t$, resulting in non-confident predictions (i.e., the model does not favor a single class, such as the zone of $k = 2, 3, ...$ in Eq. 6), EM alone for prediction $\mathbf{p}(\mathbf{x}_t)$ is insufficient to achieve zero-entropy convergence. This is especially challenging in the context of TTA, where the model observes the data once and updates it once during inference [34].

In other words, full convergence requires a well-defined target and appropriate supervision. To this end, leveraging the model's discriminative power [11], we treat the most probable class from the Softmax as the target class. We then supervise the model with cross-entropy. The Targeted Class Convergence (TCC) loss is defined as:

$$\ell_{TCC}(\theta) = \frac{1}{|\mathcal{B}_t|} \sum_{\mathbf{x}_t \in \mathcal{B}_t} \left[ -\log \left( \frac{\exp(f_\theta(\mathbf{x}_t)[\tilde{y}])}{\sum_k \exp(f_\theta(\mathbf{x}_t)[k])} \right) \right], \tag{16}$$

where $\tilde{y} = \arg\max_k p(y = k|\mathbf{x}_t)$ is the target class. Eq. 16 defines our second objective.

### 4.3 Overall Objective for Test-Time Adaptation

The total loss for a novel, entropy- and energy-based TTA approach, ReTTA, is defined as the combination of $\ell_{SSM}(\theta)$, $\ell_{TCC}(\theta)$, and the EM loss $\ell_{EM}(\theta)$ as follows:

$$\ell_{ReTTA}(\theta) = \ell_{EM}(\theta) + \lambda_1(\alpha) \cdot \ell_{SSM}(\theta) + \lambda_2 \cdot \ell_{TCC}(\theta), \tag{17}$$

where $\ell_{EM}(\theta)$ is defined in Eq. 4 for the mini-batch $\mathcal{B}_t$, and $\lambda_1$ and $\lambda_2$ are the respective coefficients. In practice, $\ell_{SSM}(\theta)$ varies across different domains (due to various types of covariate shifts [14]) and mini-batches. Thus, balancing it with $\ell_{EM}(\theta)$ is crucial for each adaptation. However, as the domain and data are unpredictable at test time, we propose a self-adjusting balancing method.

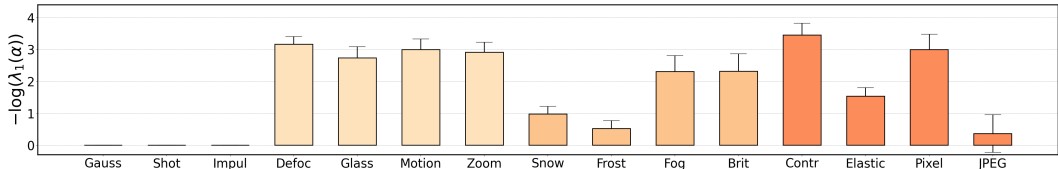

Figure 2: Breakdown of the self-adjusting coefficient $\lambda_1$ during total TTA iterations on ImageNet-C (severity 5), based on Table 1. The negative-log scale has zero corresponding to $\lambda_1 = 1$, and higher values indicate near-zero $\lambda_1$. The four colors represent Noise, Blur, Weather, and Digital groups.

**Self-adjusting coefficient.** To achieve the seemingly challenging goal of self-adjusting the balance between $\ell_{EM}(\theta)$ and $\ell_{SSM}(\theta)$, we leverage the concept of multi-objective optimization [29]. Given two objectives, the optimization problem is formulated as $\min_{\alpha \in [0,1]} ||\alpha \nabla_\theta \ell_{EM}(\theta) + (1 - \alpha) \nabla_\theta \ell_{SSM}(\theta)||_2^2$, a quadratic function of $\alpha$. The analytical solution is then given by:

$$\alpha = \frac{(\nabla_\theta \ell_{SSM}(\theta) - \nabla_\theta \ell_{EM}(\theta))^T \nabla_\theta \ell_{SSM}(\theta)}{||\nabla_\theta \ell_{EM}(\theta) - \nabla_\theta \ell_{SSM}(\theta)||_2^2}. \tag{18}$$

To ensure that the effect of $\ell_{EM}(\theta)$ is preserved while still utilizing $\ell_{SSM}(\theta)$ relatively, we clip $\alpha$ using the function $\lambda_1(\alpha) = \max(\min((1 - \alpha)/\alpha, 1), 0)$, keeping then $\lambda_1$ remains within a practical range. This adjustment maintains stability in balancing across diverse covariate shifts. In Section 5, we evaluate the effectiveness and versatility of ReTTA on various covariate shifts.

## 5    Experiment

We conduct experiments to validate the following: (1) the performance of ReTTA compared to existing entropy- and energy-based TTA methods under various distribution shifts, including challenging scenarios such as online label shifts; (2) the self-adjusting impact of $\lambda_1$ within the newly introduced loss $\ell_{SSM}(\theta)$, its projection distributions, and replacing alternative losses with SSM; and (3) the contribution of $\ell_{TCC}(\theta)$ to performance, its role in reducing entropy, and the sensitivity to $\lambda_2$.

**Dataset and baseline methods.** We evaluate ReTTA on ImageNet-C [14], a widely-used benchmark for assessing model generalization under diverse distribution shifts. The dataset consists of 15 corruption types, divided into four main categories (Noise, Blur, Weather, and Digital), each with five severity levels, for a total of 1K classes. We compare ReTTA with state-of-the-art methods including entropy-based approaches MEMO [40], Tent [34], EATA [26], SAR [27], DeYO [20], and energy-based methods TEA [39] and AEA [3].

**DNN models and experimental settings.** We perform experiments using two model architectures — ResNet-50 (with BN/GN) and VitBase (with LN) — from `torchvision` and `timm`, respectively. Following SAR [27], we use SGD with momentum 0.9, a batch size of 64, and learning rates of 0.00025 (ResNet) and 0.001 (Vit). Unless otherwise stated, we also apply the data sampling and loss-reweighting scheme from DeYO [20]. For TTA, we update only the affine parameters $\theta_{\text{affine}} \subset \theta$ of the normalization layers–batch/group norm in ResNet-50 and layer norm in VitBase–following Tent [34]. Unless otherwise stated, we fix the TCC loss coefficient at $\lambda_2 = 1$. All experiments use one-shot TTA: each test sample is observed and updated once. Further hyperparameters and implementation details are provided in Appendix B.

### 5.1    Robustness to Corruption in Test Data

**Comparison on mild scenario.** Table 1 compares the performance of ReTTA with state-of-the-art entropy-based methods (MEMO, Tent, EATA, SAR, DeYO) and energy-based methods (TEA, AEA) on ImageNet-C under mild corruption conditions (severity level 5). ReTTA achieves the highest accuracy across nearly all corruption categories, outperforming the state-of-the-art method, DeYO, in challenging noise corruptions: Gaussian (+1.7%), Shot (+1.8%), and Impulse (+1.8%). ReTTA also sets new accuracy benchmarks in the Blur, Weather, and Digital categories, significantly improving complex corruption such as Contrast (+1.5%) and Motion Blur (+0.8%). Overall, ReTTA achieves an average accuracy of 49.2%, surpassing all compared methods by at least 0.6%, demonstrating robust and broad applicability under mild distribution shifts.

| Mild | Noise | | | Blur | | | | Weather | | | | Digital | | | | Avg. |
|---|---|---|---|---|---|---|---|---|---|---|---|---|---|---|---|---|
| | Gauss. | Shot | Impul. | Defoc. | Glass | Motion | Zoom | Snow | Frost | Fog | Brit. | Contr. | Elastic | Pixel | JPEG | |
| ResNet-50 (BN) | 2.2 | 2.9 | 1.8 | 17.9 | 9.8 | 14.8 | 22.5 | 16.9 | 23.3 | 24.4 | 58.9 | 5.4 | 16.9 | 20.7 | 31.7 | 18.0 |
| MEMO | 7.5 | 8.8 | 8.9 | 19.8 | 13 | 20.7 | 27.7 | 25.3 | 28.7 | 32.2 | 61.0 | 11.0 | 23.8 | 33.0 | 37.6 | 23.9 |
| Tent | 29.2 | 31.2 | 30.1 | 28.1 | 27.7 | 41.4 | 49.4 | 47.2 | 41.5 | 57.7 | 67.4 | 29.2 | 54.8 | 58.5 | 52.4 | 43.1 |
| EATA | 34.9 | 37.1 | 35.8 | 33.4 | 33.0 | 47.1 | 52.7 | 51.6 | 45.7 | 60.0 | 68.1 | 44.4 | 57.9 | 60.6 | 55.1 | 47.8 |
| SAR | 30.6 | 30.6 | 31.3 | 28.5 | 28.5 | 41.9 | 49.4 | 47.1 | 42.2 | 57.5 | 67.3 | 37.8 | 54.6 | 58.4 | 52.1 | 43.9 |
| DeYO | 35.6 | 37.9 | 37.1 | 33.8 | 34.1 | 48.5 | 52.8 | 52.7 | **46.4** | 60.6 | 68.0 | 46.1 | 58.4 | 61.5 | 55.7 | 48.6 |
| TEA* | 16.8 | 17.5 | 17.5 | 15.8 | 16.0 | 27.3 | 39.9 | 35.3 | 33.9 | 49.0 | 65.7 | 17.9 | 45.1 | 50.2 | 41.3 | 32.6 |
| AEA | 26.2 | 26.8 | 27.3 | 24.2 | 20.8 | 40.3 | 48.1 | 47.3 | 41.4 | 56.0 | 65.7 | 9.5 | 53.4 | 56.7 | 49.5 | 39.5 |
| **ReTTA (ours)** | **37.3**$_{\pm0.0}$ | **39.7**$_{\pm0.2}$ | **38.9**$_{\pm0.2}$ | **34.5**$_{\pm0.3}$ | **34.1**$_{\pm0.0}$ | **49.3**$_{\pm0.2}$ | **53.1**$_{\pm0.2}$ | **52.7**$_{\pm0.1}$ | 46.1$_{\pm0.1}$ | **60.7**$_{\pm0.1}$ | **68.2**$_{\pm0.1}$ | **47.6**$_{\pm0.3}$ | **58.6**$_{\pm0.0}$ | **61.5**$_{\pm0.0}$ | **56.0**$_{\pm0.0}$ | **49.2**$_{\pm0.0}$ |

Table 1: Comparisons with baseline TTA methods on ImageNet-C at severity level 5 under mild scenario in terms of accuracy (%). * TEA was not publicly reported and was tested directly.

| Label Shifts | Noise | | | Blur | | | | Weather | | | | Digital | | | | Avg. |
|---|---|---|---|---|---|---|---|---|---|---|---|---|---|---|---|---|
| | Gauss. | Shot | Impul. | Defoc. | Glass | Motion | Zoom | Snow | Frost | Fog | Brit. | Contr. | Elastic | Pixel | JPEG | |
| ResNet-50 (GN) | 17.9 | 19.9 | 17.9 | 19.7 | 11.3 | 21.3 | 24.9 | 40.4 | 47.4 | 33.6 | 69.3 | 36.3 | 18.7 | 28.4 | 52.2 | 30.6 |
| MEMO | 18.4 | 20.6 | 18.4 | 17.1 | 12.7 | 21.8 | 26.9 | 40.7 | 46.9 | 34.8 | 69.6 | 36.4 | 19.2 | 32.2 | 53.4 | 31.3 |
| Tent | 3.6 | 4.2 | 4.4 | 16.5 | 5.9 | 26.9 | 28.4 | 17.9 | 26.2 | 2.3 | 72.2 | 46.1 | 7.3 | 52.3 | 56.2 | 24.7 |
| EATA | 25.7 | 28.6 | 24.8 | 18.5 | 19.6 | 24.1 | 28.4 | 35.3 | 33.0 | 41.2 | 65.2 | 33.3 | 28.0 | 42.4 | 43.1 | 32.7 |
| SAR | 33.7 | 36.9 | 35.3 | 19.3 | 20.3 | 33.8 | 29.8 | 21.9 | 44.7 | 34.9 | 71.9 | 46.7 | 6.6 | 52.3 | 56.2 | 36.3 |
| DeYO | 42.5 | 44.9 | 43.8 | 22.2 | 16.3 | 41.0 | 13.2 | 52.2 | **51.5** | 39.7 | 73.4 | **52.6** | 46.9 | 59.3 | 59.3 | 43.9 |
| TEA* | 0.4 | 0.4 | 0.4 | 0.2 | 0.1 | 0.4 | 1.2 | 1.1 | 1.3 | 0.4 | 13.5 | 0.5 | 0.3 | 0.3 | 5.0 | 1.7 |
| **ReTTA (ours)** | **42.7**$_{\pm0.3}$ | **45.1**$_{\pm0.1}$ | **44.2**$_{\pm0.2}$ | **29.4**$_{\pm2.5}$ | **22.9**$_{\pm5.8}$ | **41.1**$_{\pm0.1}$ | **34.4**$_{\pm14.4}$ | **52.8**$_{\pm0.5}$ | 51.1$_{\pm0.1}$ | **58.5**$_{\pm0.2}$ | **73.5**$_{\pm0.1}$ | 49.8$_{\pm0.2}$ | **48.4**$_{\pm0.7}$ | **59.8**$_{\pm0.3}$ | **59.3**$_{\pm0.0}$ | **47.5**$_{\pm0.4}$ |
| VitBase (LN) | 9.4 | 6.7 | 8.3 | 29.1 | 23.4 | 34.0 | 27.1 | 15.8 | 26.4 | 47.4 | 54.7 | 44.0 | 30.5 | 44.5 | 47.6 | 29.9 |
| MEMO | 21.6 | 17.4 | 20.6 | 37.1 | 29.6 | 40.6 | 34.4 | 25.0 | 34.8 | 55.2 | 65.0 | 54.9 | 37.4 | 55.5 | 57.7 | 39.1 |
| Tent | 33.9 | 1.8 | 27.2 | 54.8 | 52.9 | 58.6 | 54.3 | 12.4 | 11.7 | 69.7 | 76.3 | 66.3 | 59.6 | 69.7 | 66.6 | 47.7 |
| EATA | 36.2 | 34.7 | 35.5 | 43.4 | 44.3 | 49.3 | 48.5 | 53.2 | 53.5 | 62.3 | 72.7 | 18.8 | 58.0 | 64.7 | 62.8 | 49.2 |
| SAR | 42.3 | 34.9 | 44.1 | 50.0 | 50.5 | 55.6 | 53.1 | 59.7 | 47.2 | 66.2 | 75.2 | 50.3 | 60.1 | 67.3 | 65.0 | 54.8 |
| DeYO | 53.5 | 36.0 | 54.6 | 57.6 | 58.7 | 63.7 | 46.2 | 67.6 | 66.0 | **73.2** | 73.4 | 66.7 | 69.0 | 73.5 | 70.3 | 62.3 |
| TEA* | 6.9 | 13.2 | 14.6 | 0.9 | 1.4 | 7.1 | 3.1 | 0.6 | 1.4 | 66.9 | 73.7 | 62.1 | 1.4 | 68.2 | 63.8 | 25.7 |
| **ReTTA (ours)** | **54.0**$_{\pm0.1}$ | **55.0**$_{\pm0.1}$ | **55.2**$_{\pm0.1}$ | **57.8**$_{\pm0.2}$ | **58.7**$_{\pm0.2}$ | **64.7**$_{\pm0.1}$ | **58.5**$_{\pm7.5}$ | **69.0**$_{\pm0.4}$ | **67.1**$_{\pm0.1}$ | 71.2$_{\pm0.2}$ | **77.9**$_{\pm0.0}$ | **67.6**$_{\pm1.0}$ | **69.8**$_{\pm0.4}$ | **74.1**$_{\pm0.2}$ | **71.6**$_{\pm0.3}$ | **64.8**$_{\pm0.5}$ |

Table 2: Comparisons with baseline TTA methods on ImageNet-C (severity 5) under online label shifts (imbalance ratio=∞) in accuracy (%). * TEA was not publicly reported and was tested directly.

In this evaluation, Figure 2 illustrates the impact of the self-adjusting coefficient $\lambda_1$ on corruption-specific performance when applying SSM. The influence of SSM is most pronounced in the Noise category, where it is fully utilized and also shows to be essential in challenging corruptions like Frost and JPEG. Interestingly, in a more difficult corruption case, such as the Contrast, we observe that SSM is used conservatively, allowing the TCC to adjust the impact of EM principally. This self-adjustment signifies the importance of the adaptive balance between entropy and energy.

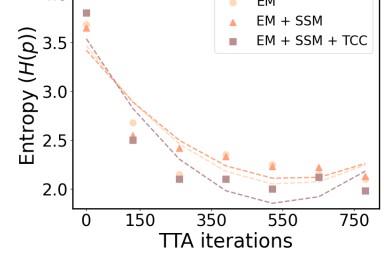

Figure 3: Effects of TCC (Defocus)

**Comparison on online label shifts.** We evaluate ReTTA under severe online label shifts, following the setting of an infinite imbalance ratio ($p_t^{max}(y)/p_t^{min}(y) = \infty$) as in SAR. Table 2 presents the results for this challenging scenario, highlighting ReTTA's robustness on both ResNet-50 (GN) and VitBase (LN). On ResNet-50 (GN), ReTTA significantly outperforms the state-of-the-art method, DeYO, across difficult corruptions, notably Defocus (+7.2%), Zoom (+21.2%), and Fog (+18.8%), achieving an overall improvement of 3.6% in average accuracy. Similarly, for VitBase (LN), ReTTA surpasses DeYO on almost all corruption categories, with notable improvements on Impulse Noise (+0.6%), Zoom Blur (+12.3%), and Pixel (+0.6%), resulting in an average accuracy gain of 2.5%. These results demonstrate ReTTA's outstanding robustness to label distribution shifts.

## 5.2 Ablation Study

**Effects of the balancing parameter $\lambda_2$.** Figure 4(a) shows how varying $\lambda_2$ impacts ReTTA's accuracy. Performance peaks at $\lambda_2 = 1$, which we adopt for all experiments; deviations in either direction degrade accuracy, showcasing the importance of a well-tuned TCC coefficient. Since TTA is unsupervised, overly large $\lambda_2$ can be detrimental. Figure 3 further demonstrates that with $\lambda_2 = 1$, ReTTA reduces entropy while boosting accuracy over EM [20] on the Defocus corruption (Table 1).

**Effects of alternatives for SSM.** Figure 4(b) shows the impact of SSM alternatives, including Score Matching (SM) and Sliced Score Matching with Variance Reduction (SSM-VR), which applies when $p(\mathbf{v})$ follows a Normal distribution [33]. While SSM, our chosen loss, outperforms SM (which shows marginal gains), it performs slightly better than SSM-VR, with an average difference of approximately 0.1%. This demonstrates SSM's effectiveness within ReTTA.

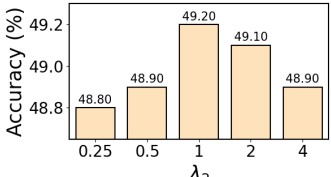
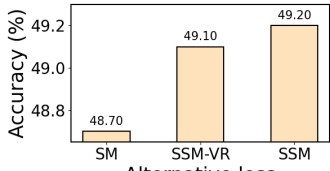
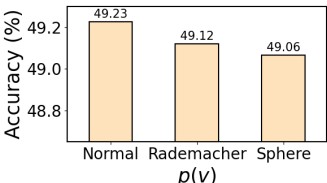

(a) Effects of varying $\lambda_2$  (b) Effects of alternatives for SSM  (c) Effects of varying $p(\mathbf{v})$

Figure 4: Effects of varying components in ReTTA: $\lambda_2$ for balancing TCC, alternative losses for SSM, and projection vector distributions within SSM. Experimental settings match those in Table 1.

| Label Shifts | Noise | | | Blur | | | | Weather | | | | Digital | | | | Avg. |
|---|---|---|---|---|---|---|---|---|---|---|---|---|---|---|---|---|
| | Gauss. | Shot | Impul. | Defoc. | Glass | Motion | Zoom | Snow | Frost | Fog | Brit. | Contr. | Elastic | Pixel | JPEG | |
| ResNet-50 (GN) + EM [20] | 42.5 | 44.9 | 43.8 | 22.2 | 16.3 | 41.0 | 13.2 | 52.2 | 51.5 | 39.7 | 73.4 | 52.6 | 46.9 | 59.3 | 59.3 | 43.9 |
| +SSM | 42.2 | 44.6 | 43.8 | 26.7 | 22.3 | 40.7 | 13.4 | 51.6 | 50.8 | 58.4 | 73.1 | 49.8 | 48.3 | 59.5 | 59.1 | 45.6 |
| +TCC | 42.5 | 45.0 | 43.4 | 22.5 | **23.4** | **41.5** | 28.7 | **53.1** | **51.5** | 59.7 | 73.6 | **52.8** | 46.3 | 59.5 | 59.5 | 46.9 |
| ↳ReTTA (Eq. 17) | **42.7** | **45.1** | **44.2** | **29.4** | 22.9 | 41.1 | **34.4** | 52.8 | 51.1 | **59.8** | 73.5 | 49.8 | **48.4** | **59.8** | **59.3** | **47.5** |
| VitBase (LN) + EM [20] | 53.5 | 36.0 | 54.6 | 57.6 | 58.7 | 63.7 | 46.2 | 67.6 | 66.0 | **73.2** | 77.9 | 66.7 | 69.0 | 73.5 | 70.3 | 62.3 |
| +SSM | **54.1** | **55.0** | **55.4** | 57.8 | 58.4 | 64.7 | **59.2** | 69.0 | 67.0 | 71.4 | 77.0 | 67.4 | 69.4 | 74.1 | 70.8 | 64.7 |
| +TCC | 53.9 | 54.8 | 55.0 | 57.8 | 58.1 | 64.4 | 41.8 | 68.1 | 66.7 | 71.1 | 77.9 | 67.1 | 69.6 | 73.9 | 69.3 | 63.3 |
| ↳ReTTA (Eq. 17) | 54.0 | 55.0 | 55.2 | **57.8** | **58.7** | **64.7** | 58.5 | **69.0** | **67.1** | 71.2 | **77.9** | **67.6** | **69.8** | **74.1** | **71.6** | **64.8** |

Table 3: Effect of components in ReTTA. Each denotes accuracy (%) on ImageNet-C (severity 5) under online label shifts (imbalance ratio = ∞), with DeYO as the baseline EM method.

| Mild | Noise | | | Blur | | | | Weather | | | | Digital | | | | Avg. |
|---|---|---|---|---|---|---|---|---|---|---|---|---|---|---|---|---|
| | Gauss. | Shot | Impul. | Defoc. | Glass | Motion | Zoom | Snow | Frost | Fog | Brit. | Contr. | Elastic | Pixel | JPEG | |
| ResNet-50 (BN) + EM [26] | 34.9 | 37.1 | 35.8 | 33.4 | 33.0 | 47.1 | 52.7 | 51.6 | 45.7 | 60.0 | 68.1 | 44.4 | 57.9 | 60.6 | 55.1 | 47.8 |
| ↳ReTTA (Eq. 17) | **35.1** | **37.8** | **36.0** | 33.7 | 33.0 | **47.5** | **52.9** | 51.7 | **45.8** | **60.1** | **68.1** | **44.7** | **57.9** | **60.7** | **55.5** | **48.0** |
| ResNet-50 (BN) + EM [27] | 30.6 | 30.6 | 31.3 | **28.5** | **28.5** | 41.9 | 49.4 | 47.1 | 42.2 | 57.5 | 67.3 | 37.8 | 54.6 | 58.4 | 52.1 | 43.9 |
| ↳ReTTA (Eq. 17) | **31.8** | **34.1** | **32.7** | 27.8 | 27.9 | **44.1** | **50.7** | **48.3** | **42.4** | **58.5** | **67.7** | **40.5** | **55.4** | **59.2** | **53.2** | **44.9** |
| ResNet-50 (BN) + EM [20] | 35.6 | 37.9 | 37.1 | 33.8 | 34.1 | 48.5 | 52.8 | 52.7 | **46.4** | 60.6 | 68.0 | 46.1 | 58.4 | 61.5 | 55.7 | 48.6 |
| ↳ReTTA (Eq. 17) | **37.3** | **39.7** | **38.9** | **34.5** | **34.1** | **49.3** | **53.1** | 52.7 | 46.1 | **60.7** | **68.2** | **47.6** | **58.6** | **61.5** | **56.0** | **49.2** |

Table 4: Adaptivity of components in ReTTA applied to state-of-the-art EM methods (EATA, SAR, and DeYO). Each denotes accuracy (%) on ImageNet-C (severity 5) under mild scenarios.

**Effects of projection distributions.** Figure 4(c) shows the effect of different projection distributions, Rademacher $\{\pm 1\}^D$ and the uniform over the hypersphere (Sphere). Normal is our chosen distribution. The minimal performance variation suggests that the choice of projection distribution has little impact on accuracy, demonstrating ReTTA's insensitivity to different projections.

**Effects of components in ReTTA.** Table 3 shows that integrating SSM improves performance, especially for ResNet-50 (GN), with additional gains from incorporating TCC. The combination of SSM and TCC outperforms the baseline EM method, demonstrating that ReTTA's integration of entropy and energy-based optimization provides a robust, general-purpose solution. This approach excels across various distribution shifts, particularly under challenging online label shifts, as also reflected in improvements with VitBase (LN).

**Effects of SSM and TCC in state-of-the-art EM methods.** Table 4 shows that combining SSM and TCC into EATA, SAR, and DeYO yields accuracy gains across most corruption types. DeYO sees its largest gain in the Noise category, while SAR outperforms its baseline on all but Defocus and Glass corruptions. EATA shows marginal gains—speculatively because its built-in forgetting mitigation dampens the impact of deviated updates from the original parameters. While gains for Defocus and Glass are more modest overall, integrating energy-based and class-targeted components in ReTTA effectively strengthens performance to diverse distribution shifts.

## 5.3 Case Study

**Potential in test-time domain adaptation.** We further evaluate ReTTA under mild adaptation conditions on three additional ImageNet-scale out-of-distribution benchmarks. ImageNet-R contains rendered versions of ImageNet objects, introducing a large domain shift. In contrast, ImageNetV2 is a closely related re-sampling of the original ImageNet distribution, while ImageNet-S consists of single-channel sketch drawings. As shown in Table 5, ReTTA improves accuracy to 47.4% (ResNet-50) and 61.7% (VitBase) on ImageNet-R, comparable to its gains on ImageNet-C. On ImageNetV2, ReTTA mitigates the performance degradation often observed with EM-based methods, which tend

| ImageNet-R | ResNet-50 (GN) | VitBase (LN) |  | ImageNetV2 | ImageNet-S |
|---|---|---|---|---|---|
| No adapt. | 40.8 | 50.9 | ResNet-50 (BN) | 63.20 | 24.09 |
| Tent | 42.8 | 55.3 | Tent | 63.07 | 30.50 |
| TEA | 7.0 | 22.9 | TEA | 57.28 | 8.87 |
| EATA | 41.9 | 51.1 | EATA | 63.14 | 35.24 |
| ↳ReTTA (Algorithm 1) | 42.4 | 52.1 | ↳ReTTA (Algorithm 1) | **63.20** | 35.24 |
| SAR | 41.9 | 51.8 | SAR | 63.06 | 31.74 |
| ↳ReTTA (Algorithm 2) | 42.3 | 52.5 | ↳ReTTA (Algorithm 2) | 63.14 | 32.16 |
| DeYO | 47.0 | 61.3 | DeYO | 62.89 | 35.83 |
| ↳ReTTA (Algorithm 3) | **47.4** | **61.7** | ↳ReTTA (Algorithm 3) | 63.06 | **35.86** |

Table 5: Comparison of ReTTA with baseline TTA methods defined by Algorithms 1-3 in Appendix B.1 on three ImageNet-scale out-of-distribution benchmarks under mild adaptation settings. Each result reports accuracy (%) on ImageNet-R (rendered objects), ImageNetV2 (re-sampled validation distribution), and ImageNet-S (sketch-style grayscale images) using ResNet-50 and VitBase.

| Mild | Noise | | | Blur | | | | Weather | | | | Digital | | | | Avg. |
|---|---|---|---|---|---|---|---|---|---|---|---|---|---|---|---|---|
| | Gauss. | Shot | Impul. | Defoc. | Glass | Motion | Zoom | Snow | Frost | Fog | Brit. | Contr. | Elastic | Pixel | JPEG | |
| EATA [26] (Source) | 73.8 | 73.5 | 73.6 | 73.7 | 73.5 | 73.8 | 74.1 | 74.3 | 74.1 | 74.6 | 74.9 | 74.1 | 74.2 | 74.3 | 74.0 | 74.0 |
| ↳ReTTA (Source) | 73.4 | 73.5 | 73.3 | 73.6 | 73.4 | 74.0 | 74.1 | 74.1 | 74.0 | 74.4 | 74.7 | 74.0 | 74.1 | 74.4 | 73.9 | 73.9 |
| EATA [26] (Lifelong) | 35.3 | 38.7 | 38.0 | 34.1 | 34.0 | 47.1 | 52.9 | 50.9 | 45.6 | 59.8 | 67.9 | 44.1 | 57.4 | 60.1 | 54.9 | 48.0 |
| ↳ReTTA (Lifelong) | 36.0 | 38.8 | 38.2 | 34.2 | 33.8 | 47.4 | 53.1 | 51.4 | 45.5 | 59.9 | 68.1 | 44.6 | 57.5 | 60.7 | 54.9 | 48.3 |

Table 6: Adaptivity of ReTTA when integrated with EATA (Algorithm 1) under the lifelong adaptation setting, following the continual adaptation strategy described in [26]. Each result represents accuracy (%) on ImageNet-C (severity 5) using ResNet-50 (BN) with a source accuracy of 76.13%. "Source" denotes the validation performance on ImageNet-1k measured during the lifelong adaptation process.

to incur uncertainty-induced penalties in similar or overlapping domains. On ImageNet-S, ReTTA maintains the top rank among baselines, although the limited grayscale information and reduced modes of distribution likely constrain the influence of SSM.

**Potential in lifelong adaptation.**    Building on Lemma 3, we speculate that SSM can also mitigate forgetting. During each adaptation step, the "negative phase" slightly raises energy for samples from the source distribution, while the "positive phase" lowers energy for newly corrupted samples. Rather than offsetting each other within a single SGLD update, these two forces may reach a balanced state that broadens the source density to cover new data without overwriting its original support. Table 6 confirms that integrating ReTTA with EATA preserves similar gains in lifelong protocols as in standard TTA (+0.2%). In particular, ReTTA+EATA achieves an average improvement of +0.3% in the lifelong setting, yielding a final source accuracy of 73.9%. These results demonstrate that ReTTA complements EM not only by enhancing adaptation under distributional shifts but also by maintaining source performance during extended lifelong adaptation.

# 6   Conclusion

This paper questions the sufficiency of minimizing entropy alone for effective TTA. We identify two key obstacles in entropy minimization under distribution shifts: the inability to estimate the test data distribution and the failure to enhance the model's discriminability further. This study shows that simultaneous entropy-energy minimization is one goal-driven approach to overcoming these problems. We propose ReTTA, which combines SSM and TCC for energy reduction and discriminability. By adaptively balancing both objectives, ReTTA offers a scalable solution to distribution shifts. Extensive experiments demonstrate that ReTTA outperforms existing entropy- or energy-based TTA methods.

## Acknowledgments and Disclosure of Funding

This research was partly supported by the IITP(Institute of Information & Communications Technology Planning & Evaluation)-ITRC(Information Technology Research Center) grant funded by the Korea government(MSIT)(IITP-2025-RS-2023-00258649, 50%), the National Research Foundation of Korea(NRF) grant funded by the Korea government(MSIT) (RS-2025-00562437, 40%), and Institute of Information & communications Technology Planning & Evaluation (IITP) grant funded by the Korea government(MSIT) (No.RS-2022-00155911, Artificial Intelligence Convergence Innovation Human Resources Development (Kyung Hee University), 10%).

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
