# OpenReview forum: "Rethinking Entropy in Test-Time Adaptation: The Missing Piece from Energy Duality"
_NeurIPS.cc/2025/Conference — NeurIPS 2025 spotlight_

### Official Review · Reviewer_34Nx · 2025-06-20

**Clarity:** 3
**Significance:** 3
**Originality:** 3
**Rating:** 5
**Confidence:** 3

**Summary:**

This paper proposes a new TTA method, called ReTTA, which balances out the widely used Entropy Minimization (EM) loss with an energy objective (Sliced Score Matching, SSM). Because entropy minimization only cares about the ratio of logits to each other, there is no incentive to increase the likelihood (lower the energy) of test inputs  In addition, Targeted Cross Entropy (TCC) is used to ensure that logits hit the 0 entropy mark, instead of hovering near it. ReTTA achieves state of the art results on ImageNet-C.

**Questions:**

Would it be possible to get more intuitive/practical explanations for the ways that the energy and TCC objectives help complement entropy? Are there specific images that are difficult for entropy based TTA to classify, but not for ReTTA?

Would it be possible to see ReTTA performance on other ImageNet-size OOD datasets?

Would it be possible to see ReTTA performance in the lifelong setting seen in EATA?

**Ethical Concerns:**

["NO or VERY MINOR ethics concerns only"]

**Final Justification:**

I found the paper to be easy to read, with a novel method that made sense. The paper provided only a select number of benchmarks at first, but with the rebuttal, the authors have demonstrated ReTTA's effectiveness across: IN-C severities 1 and 3, the continual setting used in EATA, as well as other OOD ImageNet scale datasets. My concerns have been addressed, and I have no further concerns.

**Limitations:**

No, but I don't feel like this is relevant here.

**Quality:**

3

**Strengths And Weaknesses:**

The paper overall is well-structured and clear. There is theoretical motivation supplemented by practical results.

Strengths:

ReTTA seems to significantly improve performance in the classic ImageNet-C setting. The results span architectures and can be stacked with other methods, namely EATA, SAR and DeYO.

Weaknesses:

I think more motivation for why the entropy minimization objective is not enough, and why the energy objective solves this. I can see the theoretical argument, yes, but I think having some sort of practical intuition would be very helpful. Figure 2 in [1] comes to mind; I think something along those lines, that shows the problem of minimizing entropy without minimizing energy would go a long way. In the same vein, getting some insight/intuition as to why TCC is helpful would also be interesting; when entropy is close to 0, I would imagine that the entropy loss and TCC are basically the same; if so, what does TCC do that entropy doesn't?

I think the experiments should be more comprehensive; ImageNet-C Severity 5 is good enough to get a good understanding of the method's performance, but full IN-C, or at least, levels 1, 3, along with 5 should be shown. I also think ImageNet-R, ImageNet-3DCC, ImageNet-V2 or other popular, ImageNet-scale OOD datasets could boost the robustness of the results table.

How effective is ReTTA in the lifelong setting seen in [1] (Table 2)? EATA showed that forgetting was a problem with many top TTA methods, it would be interesting to see how ReTTA handles it.



Minor:
I think it would be helpful to have an explanation of the settings "Label Shift" and "Mild" in the paper, even if they are identical to ones mentioned in other papers.
Figure 4a shows that $\lambda_2$ was chosen in based on the same experimental settings seen in Table 1. If I understand this correctly, this should not be done; parameters should be picked using the IN-C holdout set of noises, not the 15 main noises.
Table 3, the rows in the first columns are all the same (should be EATA, SAR, DeYO)

[1] Niu, Shuaicheng, et al. "Efficient test-time model adaptation without forgetting." International conference on machine learning. PMLR, 2022.

---

> ### Author Rebuttal · Authors · 2025-07-31
>
> We appreciate the valuable time and effort you took to provide constructive and positive feedback. We believe these clarifications strengthen our manuscript and look forward to favorable consideration.
>
> &nbsp;
>
> >W1&Q1. When entropy is close to 0, I would imagine that the entropy loss and TCC are the same; if so, what does TCC do that entropy doesn't? Would it be possible to get more intuitive/practical explanations for the ways that the energy and TCC objectives help complement entropy? Are there images that are difficult for entropy based TTA to classify, but not for ReTTA?
>
> We view TCC as a simple yet effective way to push the model output from merely low entropy to truly zero entropy, something that EM alone cannot ensure. Intuitively, the motivation is straightforward since zero-entropy predictions often correspond to highly confident and accurate classifications, as shown in Figs. 1(b) and 1(c). When a prediction is already at zero entropy, neither EM nor TCC makes further updates. In such cases, SSM pulls the sample deeper into the high-density, in-distribution region that EM alone cannot reach. By doing so, we encourage adaptation toward in-distribution and likely correct samples. So, for most inputs with non-zero entropies, TCC remains essential: in practice, as Table 3 shows, coupling state-of-the-art EM with TCC yields dramatic gains of 7% or 20% on corruptions like Glass or Frost, and achieves over a 1% average improvement versus the EM method (our in-house test of TCC with Tent shows a 0.9% average gain, from 43.1% to 44.0%). We also find EM often fails on images where the object is tiny or occluded, whereas ReTTA succeeds. A comprehensive analysis will require new metrics, which we plan to analyze in-depth.
>
> &nbsp;
>
> >W2. full IN-C, or at least, levels 1, 3, along with 5 should be shown.
>
> We observed that at severity level 1, all TTA methods achieved comparable average performance (with ReTTA only marginally superior), whereas as corruption severity increased (levels 3 and 5; see Table 1), the combination of energy-based and entropy-based optimizations became important for maintaining higher accuracy.
>
> A. ImageNet-C Severity 1 (ResNet-50-BN)
>
> | Method         | Gauss. | Shot  | Impul. | Defoc. | Glass | Motion | Zoom  | Snow  | Frost | Fog   | Brit. | Contr. | Elastic | Pixel | JPEG  | Avg   |
> |----------------|-------:|------:|-------:|-------:|------:|-------:|------:|------:|------:|------:|------:|-------:|--------:|------:|------:|------:|
> | No adapt       |   59.6 |  57.8 |   48.4 |   59.0 |  53.7 |   64.5 |  52.2 |  54.3 |  61.1 |  61.4 |  73.8 |   64.4 |    66.4 |  63.9 |  65.9 | 60.4 |
> | Tent           |   68.0 |  67.6 |   64.3 |   66.4 |  67.3 |   70.5 |  65.4 |  65.1 |  66.4 |  70.2 |  **74.6** |   72.0 |    69.9 |  72.0 |  70.1 | 68.7 |
> | EATA           |   68.7 |  68.4 |   65.6 |   67.0 |  68.0 |   70.6 |  66.2 |  66.5 |  66.9 |  70.8 |  74.4 |   72.3 |    70.2 |  72.3 |  70.4 | 69.2 |
> | SAR            |   67.9 |  67.5 |   64.3 |   66.3 |  67.3 |   70.4 |  65.4 |  64.9 |  66.3 |  70.1 |  74.5 |   71.9 |    69.9 |  72.0 |  70.1 | 68.6 |
> | DeYO           |   68.9 |  68.6 |   66.0 |   67.2 |  68.0 |   70.5 |  66.3 |  **67.0** |  67.1 |  70.8 |  74.1 |   72.2 |    70.1 |  **72.3** |  70.2 | 69.3 |
> | TEA            |   59.4 |  58.9 |   54.0 |   56.8 |  58.5 |   61.6 |  54.7 |  56.9 |  57.0 |  62.4 |  66.8 |   63.6 |    60.3 |  64.0 |  61.2 | 59.7 |
> | **ReTTA**      |   **68.9** |  **68.7** |   **66.0** |   **67.4** |  **68.3** |   **70.7** |  **66.3** |  66.9 |  **67.1** |  **70.8** |  74.1 |   **72.4** |    **70.3** |  72.2 |  **70.4** |**69.4** |
>
> B. ImageNet-C Severity 3 (ResNet-50-BN)
>
> | Method         | Gauss. | Shot  | Impul. | Defoc. | Glass | Motion | Zoom  | Snow  | Frost | Fog   | Brit. | Contr. | Elastic | Pixel | JPEG  | Avg   |
> |----------------|-------:|------:|-------:|-------:|------:|-------:|------:|------:|------:|------:|------:|-------:|--------:|------:|------:|------:|
> | No adapt       |   27.6 | 25.0  |   25.2 |   37.9 |  16.9 |   37.7 |  35.2 |  35.2 |  32.1 |  46.7 |  69.6 |   46.0 |    55.6 |  46.2 |  59.3 | 39.7 |
> | Tent           |   54.8 | 54.3  |   53.7 |   49.2 |  46.5 |   58.8 |  57.7 |  55.9 |  48.6 |  65.8 |  72.1 |   67.1 |    69.4 |  67.5 |  65.9 | 59.2 |
> | EATA           |   57.0 | 56.8  |   56.2 |   52.4 |  50.2 |   61.0 |  59.7 |  58.6 |  51.3 |  67.1 |  72.2 |   68.2 |    69.9 |  68.1 |  66.7 | 61.0 |
> | SAR            |   54.6 | 54.1  |   53.5 |   49.3 |  46.3 |   58.6 |  57.6 |  55.6 |  48.6 |  65.6 |  72.0 |   67.1 |    69.3 |  67.3 |  65.7 | 59.0 |
> | DeYO           |   58.1 | 58.0  |   57.1 |   53.2 |  51.2 |   61.9 |  59.8 |  59.6 |  51.9 |  67.5 |  72.0 |   68.5 |    69.6 |  68.6 |  66.6 | 61.6 |
> | TEA            |   24.8 | 22.0  |   24.5 |   20.1 |  14.9 |   45.3 |  32.0 |  35.0 |  17.6 |  57.8 |  64.2 |   50.9 |    60.4 |  59.0 |  55.5 | 38.9 |
> | **ReTTA**      |   **58.6** | **58.7**  |  **58.0** |   **53.2** |  **51.5** |   **61.9** |  **59.9** |  **59.7** |  **52.1** |  **67.5** |  **72.3** |   **68.7** |   **69.9** |  **68.8** |  **66.9** |**61.9** |
> &nbsp;
>
> >W2&Q2. Would it be possible to see ReTTA performance on other ImageNet-size OOD datasets?
>
> We evaluated ReTTA under `mild’ settings on three additional ImageNet-scale OOD benchmarks. ImageNet-R (rendered images) exhibits a large domain gap compared to ImageNet, while ImageNetV2 overlaps the original ImageNet distribution, and ImageNet-S (sketches) consists of single-channel grayscale outlines. We were unable to include ImageNet-3DCC, as its download alone exceeded 3 days and more than 340 GB and stalled. Therefore, we substituted ImageNet-S instead.
>
> A. ImageNet R
>
> | Method         | ResNet 50 GN | ViT Base LN |
> |----------------|-------------:|------------:|
> | No Adapt.      |         40.8 |        50.9 |
> | Tent           |         42.8 |        55.3 |
> | EATA           |         41.9 |        51.1 |
> | SAR            |         41.9 |        51.8 |
> | DeYO           |         47.0 |        61.3 |
> | TEA            |          7.0 |        22.9 |
> | **ReTTA based on EATA**    |         42.4 |   52.1 |
> | **ReTTA based on SAR**     |        42.3 |    52.5 |
> | **ReTTA based on DeYO**      |    **47.4** |    **61.7** |
>
> B. ImageNetV2
>
> | Method         | ResNet 50 BN |
> |----------------|-------------:|
> | No Adapt.      |        **63.20** |
> | Tent           |        63.07 |
> | EATA           |        63.14 |
> | SAR            |        63.06 |
> | DeYO           |        62.89 |
> | TEA            |        57.28 |
> | **ReTTA based on EATA**    |        **63.20** |
> | **ReTTA based on SAR**     |        63.14 |
> | **ReTTA based on DeYO**    |        63.06 |
>
> C. ImageNet S
>
> | Method         | ResNet 50 BN |
> |----------------|-------------:|
> | No Adapt.      |        24.09 |
> | Tent           |        30.50 |
> | EATA           |        35.24 |
> | SAR            |        31.74 |
> | DeYO           |        35.83 |
> | TEA            |         8.87 |
> | **ReTTA based on EATA**    |        35.24 |
> | **ReTTA based on SAR**     |        32.16 |
> | **ReTTA based on DeYO**    |        **35.86** |
>
> On ImageNet-R, ReTTA achieves gains comparable to those on ImageNet-C. On ImageNetV2, where other methods sometimes experience a performance drop, ReTTA offsets that penalty, paralleling our findings in the text-classification experiments (Response to YKNW). On ImageNet-S, ReTTA maintains the top rank among baselines, although the limited grayscale information and then reduced modes of distribution likely constrain the impact of SSM.
>
> &nbsp;
>
> >W3&Q3. Would it be possible to see ReTTA performance in the lifelong setting seen in EATA?
>
> Looking back at Lemma 3, we could see that SSM might also help prevent forgetting. In each update, the “negative phase” gently raises energy (reducing the likelihood) for samples from the source distribution, while the “positive phase” lowers energy on the new corrupted samples. Rather than canceling out in a single SGLD iteration, these forces may reach a balance that slightly broadens the source density to cover new data without erasing its original support.
>
> Indeed, when comparing ReTTA + EATA in the standard TTA setting (Table 4), we observe a modest average gain of +0.2%, and in the lifelong protocol, an even larger boost of +0.3%. These results suggest that SSM and EATA can be complementary not only for adapting to distributional shifts but also for preserving source performance over time, as evidenced by the average accuracies of 74.0% and 73.9%, respectively.
>
>
> | Method                         | Gauss. | Shot  | Impul. | Defoc. | Glass | Motion | Zoom  | Snow  | Frost | Fog   | Brit. | Contr. | Elastic | Pixel | JPEG  | Avg   |
> |--------------------------------|-------:|------:|-------:|-------:|------:|-------:|------:|------:|------:|------:|------:|-------:|--------:|------:|------:|------:|
> | **EATA (Continual)**           |   35.3 |  38.7 |   38.0 |   34.1 |  **34.0** |   47.1 |  52.9 |  50.9 |  **45.6** |  59.8 |  67.9 |   44.1 |    57.4 |  60.1 |  54.9 | 48.0 |
> | **Source**          |   73.8 |  73.5 |   73.6 |   73.7 |  73.5 |   73.8 |  74.1 |  74.3 |  74.1 |  74.6 |  74.9 |   74.1 |    74.2 |  74.3 |  74.0 | **74.0** |
> | **ReTTA + EATA (Continual)**   |   **36.0** |  **38.8** |  **38.2** |   **34.2** |  33.8 |  **47.4** |  **53.1** |  **51.4** |  45.5 |  **59.9** |  **68.1** |  **44.6** |  **57.5** |  **60.7** |  **54.9** | **48.3** |
> | **Source**          |   73.4 |  73.5 |   73.3 |   73.6 |  73.4 |   74.0 |  74.1 |  74.1 |  74.0 |  74.4 |  74.7 |   74.0 |    74.1 |  74.4 |  73.9 | 73.9 |
>
> &nbsp;
>
> >Minor weaknesses
>
> Thank you for your careful reading and helpful suggestions. To address your remaining concerns, we will (1) add brief descriptions of the “Label Shift” and “Mild” settings; (2) due to the volume of experiments, we will reserve a small holdout subset of ImageNet-C corruptions for exploring $\lambda_2$, and (3) update Table 3 to include the missing EATA and SAR entries.

---

> > ### Comment · Reviewer_34Nx · 2025-08-01
> >
> > Thanks for your answers - my initial concerns were addressed, and I have increased my score. I think the the results discussed should be included in the final paper, as well as any other OOD ImageNet scale benchmarks, as they help show ReTTA's effectiveness in many settings.

---

> ### Author Response · Authors · 2025-08-01
>
> Thank you for your valuable and encouraging feedback. We truly appreciate your recognition of our work. As suggested, we will include the discussed results, including those on OOD ImageNet scale benchmarks, in the final version to better demonstrate ReTTA’s effectiveness across diverse settings.
>
> &nbsp;
>
> Sincerely,
>
> The Authors

---

### Official Review · Reviewer_oKyZ · 2025-06-26

**Clarity:** 3
**Significance:** 2
**Originality:** 3
**Rating:** 5
**Confidence:** 3

**Summary:**

Many existing test-time adaptation (TTA) methods are based on entropy minimization (EM), which encourages confident predictions on unlabeled test data. This paper revisits EM through its duality with energy, and argues that minimizing entropy alone is insufficient for TTA. The authors propose ReTTA, a unified framework that supplements EM with two additional objectives: (1) a sampling-free energy minimization objective via Sliced Score Matching (SSM), and (2) a Targeted Class Convergence (TCC) loss to promote discriminative predictions. Experiments on ImageNet-C demonstrate the effectiveness of ReTTA over existing EM-based and energy-based baselines.

**Questions:**

1. Figure 1 is not very clear to me, where I can't see much difference between Figure 1 (b) and (c). Are the authors suggesting that ReTTA (c) provides a clear improvement over EM (b)? If so, it would be helpful to highlight or quantify this difference more explicitly. Additionally, for (d), it would improve clarity to include a legend or annotation explaining the meanings of K and Z, so the figure is more self-contained.
2. The paper claims ReTTA is a “scalable solution” to distribution shifts in Conclusion. What exactly does “scalable” mean here, in terms of data size, model size, or corruption severity? Is there any empirical evidence supporting this?
3. I have doubt on the impact of TCC loss. Since test-time data is corrupted, the top-1 prediction may be wrong in some cases. Using it as a pseudo-target could mislead the model, especially in low-confidence cases. Table 3 suggests that TCC offers limited benefit when used alone. Could you provide more analysis on when and why TCC is effective?

**Ethical Concerns:**

["NO or VERY MINOR ethics concerns only"]

**Final Justification:**

After reading the rebuttal and other reviews, all my concerns have been satisfactorily resolved. The authors further provide much additional experiments in the rebuttal to better empirically support the claim. So I have increased my score to vote for the acceptance.

**Limitations:**

The authors mentioned that the following in the checklist regarding the Limitations:

Answer: [Yes]. Justification: Limitations are discussed in the Appendix.

But it seems that I cannot access to the appendix...
I suggest the authors add this part.

**Paper Formatting Concerns:**

None.

**Quality:**

3

**Strengths And Weaknesses:**

- Strengths
1. The motivation of the proposed ReTTA method is welled analyzed. The insights of using both entropy and energy is valuable and novel.
2. ReTTA’s components (SSM and TCC) are easily pluggable into existing EM-based TTA methods, as demonstrated in the ablation studies (Table 4).

- Weaknesses
1. In the standard TTA setting (Table 1), ReTTA’s improvements over strong baselines like DeYO are relatively small (+0.6% average). The empirical results does not show a very significant advantage of using ReTTA, especially on mild scenario.
2. While the method is motivated by covariate shift, the experimental setup includes a label shift, making it unclear what shift assumptions ReTTA is designed for. Given the shift in both p(x) and p(y), it is hard to say if this is still covariate shift or label shift since it's hard to maintain p(y|x) or p(x|y) unchanged.

---

> ### Author Rebuttal · Authors · 2025-07-31
>
> We appreciate the valuable time and effort you took to provide constructive and positive feedback. We believe these clarifications strengthen our manuscript and look forward to favorable consideration.
>
> &nbsp;
>
> >W1. In the standard TTA setting (Table 1), ReTTA’s improvements over strong baselines like DeYO are relatively small (+0.6% average). The empirical results does not show a very significant advantage of using ReTTA, especially on mild scenario.
> >>Q1. Figure 1 is not very clear to me, where I can't see much difference between Figure 1 (b) and (c). Are the authors suggesting that ReTTA (c) provides a clear improvement over EM (b)? If so, it would be helpful to highlight or quantify this difference more explicitly. Additionally, for (d), it would improve clarity to include a legend or annotation explaining the meanings of K and Z, so the figure is more self-contained.
>
> Thank you for your careful reading and observations. It is worth noting that even on mild shifts, leading TTA methods yield only modest gains. For instance, DeYO improves over EATA by +0.8% on average (Table 1), so ReTTA’s +0.6% over DeYO is almost in line with the state of the art. When applied to SAR, ReTTA yields a +2.7% net gain in accuracy (Table 4), corresponding to roughly 1,350 additional correct predictions out of 50,000. Although adaptation does flip a few accurate predictions to incorrect and vice versa, the dominant effect is this net gain of 1,350 samples. This improvement explains why Figs. 1(b) and 1(c) appear visually subtle. To clarify, we will annotate the figure with the number of samples corrected and add a legend for Fig. 1(d) (refer to the caption of Fig. 2 in the Appendix for an example).
>
> &nbsp;
>
> >W2. While the method is motivated by covariate shift, the experimental setup includes a label shift, making it unclear what shift assumptions ReTTA is designed for. Given the shift in both p(x) and p(y), it is hard to say if this is still covariate shift or label shift since it's hard to maintain p(y|x) or p(x|y) unchanged.
>
> Although the experimental protocols include both mild (covariate) and label shifts, this diversity cannot alter ReTTA’s core motivation: adapting to covariate shift in $p(x)$. In the mild setting, corruptions modify $p(x)$ under a uniform label distribution $p(y)$, so the resulting shift in $p(x|y)$ remains interpretable as a covariate shift. In label-shift tests, we replace the uniform prior with a skewed distribution $q(y)$, yielding $q(x)=\sum_y p(x|y)q(y)$, which again manifests as a new covariate shift in the marginal $q(x)$. Thus, regardless of stress-testing under combined shifts, ReTTA’s design and motivation remain focused on adaptation for covariate shifts.
>
> &nbsp;
>
> >Q2. The paper claims ReTTA is a “scalable solution” to distribution shifts in Conclusion. What exactly does “scalable” mean here, in terms of data size, model size, or corruption severity? Is there any empirical evidence supporting this?
>
> We intended to highlight ReTTA’s ability to adapt to diverse distribution shifts, made possible by its self-adjusting coefficient, since this mechanism enables the method to balance likelihood and entropy signals on the fly without the need for manual tuning across different distribution shifts. Now, in response to Reviewer GM3k, YKNW, and 34Nx, we have further evaluated ReTTA across ImageNet-C severities (1 and 3), various Out-of-Distribution domains (ImageNet-R, V2, S), and even text classification tasks. We believe that these results can also support our argument that ReTTA scales across different domain types, modalities, and shift intensities.
>
> &nbsp;
>
> >Q3. I have doubt on the impact of TCC loss. Since test-time data is corrupted, the top-1 prediction may be wrong in some cases. Using it as a pseudo-target could mislead the model, especially in low-confidence cases. Table 3 suggests that TCC offers limited benefit when used alone. Could you provide more analysis on when and why TCC is effective?
>
> Thank you for raising this important point. Our motivation behind TCC is to amplify the effect of entropy minimization (EM) by explicitly targeting samples near truly zero entropy where correctness is more likely, as shown in our empirical observations. TCC offers a straightforward and effective method for pushing predictions into this regime. However, we also recognize that such aggressive guidance may break the conservative nature of EM and could be vulnerable to errors if applied indiscriminately.
>
> To mitigate this, as shown in Algorithms 1–3, we employ strong entropy-based data filtering —a strategy widely used in state-of-the-art EM approaches —to ensure that TCC is applied only when confidence is sufficiently high. In this design, TCC is not intended to function alone, but as a complementary module within EM pipelines that already incorporate safety mechanisms.
>
> Thus, Table 3 should not be interpreted as showing TCC’s limited effect in isolation, but rather as evidence of its effectiveness under proper safeguards. In fact, when combined with EM, TCC delivers dramatic improvements of +7% and +20% on corruptions like Glass and Frost, and an average gain of over +1% compared to the EM method.

---

> > ### Comment · Reviewer_oKyZ · 2025-08-04
> >
> > Thank the authors for the detailed responses! After reading the rebuttal and other reviews, all my concerns have been satisfactorily resolved. I will increase my score accordingly.

---

> > > ### Author Response · Authors · 2025-08-04
> > >
> > > Thank you for your valuable and encouraging feedback. We are glad to hear that our responses contributed to addressing your concerns. We truly appreciate your recognition of our work.
> > >
> > > &nbsp;
> > >
> > > Sincerely,
> > >
> > > The Authors

---

### Official Review · Reviewer_YKNW · 2025-07-03

**Clarity:** 4
**Significance:** 3
**Originality:** 3
**Rating:** 5
**Confidence:** 3

**Summary:**

The paper presents a test time adaptation framework called ReTTA that focuses on joint entropy and energy minimization. For this, it first studies the dual relationship between entropy and energy and shows that minimization of entropy alone does not ensure minimization of energy (or maximization of likelihood of the test data). Hence, both need to be explicitly minimized. Based on this, they propose ReTTA’s objective: EM (entropy minimization) + SSM (energy minimization) + TCC (for driving entropy to zero).

For evaluation, they consider ImageNet-C and two neural network architectures: ResNet-50 and ViT. The results show that their entropy+energy based approach outperforms SOTA entropy-only or energy-only based approaches.

**Questions:**

This paper presents results on only ImageNet-C. However, a list of datasets for TTA experimentation (covering various domains and problems) can be found at https://docs.google.com/spreadsheets/d/10tOlFDA5hLSpyv5Wv8zRcXSbUEDLfxP-YhU82AZvYJo/edit?pli=1&gid=0#gid=0
1. How well will ReTTA generalize to other datasets?
2. Is the method directly applicable to other domains such as sequential text or speech?
3. If no, will it require domain or problem-specific modifications?

**Minor comments:**
1. Fig 1d is not described in the caption.
2. In Table 1, same highest values for DeYO are not in bold. Similar observation in Table 3 and Table 4.
3. It might be helpful for the reader if entropy minimization and energy minimization could be differentiated as EntM and EnM, respectively. I had to look up multiple times whether EM referred to entropy or energy minimization.

**Ethical Concerns:**

["NO or VERY MINOR ethics concerns only"]

**Final Justification:**

Based on my reading and other reviews, I found the proposed method to be novel and useful for the research community. The motivation is theoretically grounded and the effectiveness of the method has been verified empirically. My main concern was related to the method's generalization to other image-based datasets, which the authors have addressed with additional experiments during the rebuttal. The paper is overall well written and clear to read as well. I therefore recommend an "Accept" for this paper.

**Limitations:**

Yes

**Quality:**

4

**Strengths And Weaknesses:**

**Strengths:**
1. Originality: the paper highlights the need to minimize both entropy and energy for TTA, providing new insights on this topic. Previous methods have either focused on entropy minimization or energy minimization but not on both together. Based on this, the paper presents a new objective function that combines entropy and energy minimization for TTA, including an auto-adjusting coefficient to balance the two objectives.
2. Significance: the results are important for the research community because, based on the insights and results presented in this study, researchers will likely focus more on the joint minimization of entropy + energy, possibly deriving new methods for TTA.
3. Quality: the paper is technically good, with claims and motivation for the proposed objective function well supported by theoretical (Lemma 1 and Lemma 2, Theorem 1) and experimental analysis (Fig 1d) on the relationship of entropy and energy minimization. Although shown for only one dataset, experimental results show the effectiveness of the proposed objective under various distribution shifts, including the extreme case of online label shifts. The study is comprehensive covering additional aspects such as computational overhead, impact of hyperparameters and their sensitivity. Limitation of the proposed method (for the streaming case) has also been discussed.
4. Clarity: the paper is well written and clearly organized, with the required flow and necessary information needed for a reader’s clear understanding. Authors do a very good job of presenting the research story starting with the motivation (derived for the study of relationship between entropy and energy), followed by the proposal of different components of the objective function.

**Weaknesses:**
1. Unlike baseline methods, such as Tent (3-4 datasets), EATA (3 datasets), DeYO (5 datasets), this paper presents experiments on only ImageNet-C. There is no study or discussion on how well ReTTA will generalize to other datasets and domains (text, speech).

---

> ### Author Rebuttal · Authors · 2025-07-31
>
> We appreciate your valuable time and effort in providing constructive and positive feedback. We have clarified the remaining points on Q1-Q3 below and added responses and a preliminary case study in text to show how ReTTA can extend beyond images.
>
> &nbsp;
>
> >Q1. Generalization to other image datasets]
>
> ReTTA’s three pillars, Entropy Minimization (EM), Sliced Score Matching (SSM), and Targeted Class Convergence (TCC), are fully modular and plug-and-play for image classification. In practice, once a model (e.g., CNN or ViT) is trained:
> 1. Entropy: We can compute the entropy loss $-H(p)$ on images and take a gradient step in model parameters.
> 2. SSM: We only need the pixel-space gradient, which any differentiable model naturally provides.
> 3. TCC: We then apply a cross-entropy push toward the most probable class.
>
> Therefore, no extra parameters and no architectural tweaks are required to move from ImageNet-C to variants like ImageNetV2, ImageNet-R, ImageNet-S, or other image datasets. We covered evaluations on these ImageNet variant datasets in response to Reviewer 34Nx.
>
> &nbsp;
>
> >Q2 and Q3. Applicability to text or speech modalities
>
> While pixel-space gradients make SSM straightforward for images, raw text and audio waveforms lack such a grid. Directly computing score and Hessian on token IDs or raw waveforms may not capture meaningful structure. Instead, one should:
> 1. Choose a representation (e.g., token embeddings, spectrogram frames).
> 2. Compute then the Jacobian and Hessian trace in that representation space.
> 3. Formulate SSM based on those derivatives.
>
> To show how this works in text, we conducted an exploratory sentiment classification study on Amazon Polarity Reviews.
>
> &nbsp;
>
> A. Data, Model, and Task
>
> We evaluated ReTTA on a cross-domain text classification scenario using the Amazon Polarity Reviews dataset. The source domain was Books, and the target domain was Electronics. Each review carried a 1–5 star rating; we binarized these into negative (ratings 1–2 $\rightarrow$ label 0) and positive (ratings 4-5 $\rightarrow$ label 1), discarding neutral reviews with rating 3. At test time, we deployed a Books-pretrained model to classify Electronics reviews under the TTA paradigm, using DistilBERT [1] from HuggingFace.
>
> &nbsp;
>
> B. Implementation of SSM in Embedding Space
>
> To extend SSM to text, we treated each input as the token-embedding sequence $s\in \mathbb{R}^{l\times 768}$ ($l$=sequence length) produced by DistilBERT’s encoder. For each $s$, we computed the Jacobian $\nabla_{s}\log p(s)$ and Hessian $\nabla^2_{s}\log p(s)$ via differentiation with the energy-based models with respect to $s$, i.e., $p(s):=\exp(-E_\theta(s))/Z(\theta)$. We then sampled random projection vectors $v\in\mathbb{R}^{l\times 768}$ and minimized the SSM loss:
>
> $\mathbb{E}_{p(v)}\left[ v^\top \nabla^2_s \log  p(s) v + \frac{1}{2} || v^\top \nabla_s \log  p(s) ||^2 \right]$.
>
> &nbsp;
>
> C. Experimental settings for text classification
>
> We used the AdamW with a base learning rate scaled linearly by batch size, specifically setting $2e-5\times(\text{batch size} / 32)$ and a batch size of 256. The training epoch was set to 3. For Books training, we retrained all transformer layers and the classifier heads, keeping embeddings frozen. For Electronics TTA, we only updated the LayerNorm affine weights in transformer layers. Sequences were truncated/padded to 128 tokens; Books were split 90%/10% train/validation. Given that we addressed a single source to target shift, we did not employ a self‑adjusting coefficient; instead, we fixed both $\lambda_1$ and $\lambda_2$ to 1.
>
> &nbsp;
>
> D. Preliminary Experimental Results
>
> |Dataset | Method                           | Accuracy (%) |
> |--------|-------------------------------------|-------------:|
> | **Books** (Source) |        -          |        97.21 |
> |**Electronics** (Target) |  **No adapt.**     |        **92.99** |
> | | **Tent** (EM)                  |        **87.73** |
> | | **+Data‑filter (entropy < 0.2)**  |        87.75 |
> | | **+SSM**                        |        92.31 |
> | | **+TCC**                        |        87.76 |
> | | **ReTTA** (EM+SSM+TCC)    |        **92.28** |
>
> The DistilBERT model achieved 97.21% accuracy on the Books (source) domain. Without adaptation, the performance drops to 92.99% on Electronics (target). The results imply that the positive/negative sentiments from Books to Electronics highly overlap.
>
> * EM only (Tent) fell to 87.73%, showing that naive EM can be harmful under mild shifts.
>
> * Data filtering (entropy <0.2), adaptation with only those examples, yielded a marginal gain (87.75%) over EM.
>
> * EM with SSM recovered much of the gap, boosting adaptation accuracy to 92.31% on Electronics.
>
> * EM with TCC improved to 87.76%, which is marginally better than EM alone.
>
> * ReTTA (EM + SSM + TCC) achieved 92.28%, comparable to SSM only but markedly better than EM-only.
>
> These findings suggest that when the target distribution is not too dispersed, such as in the case of Books (source) and Electronics (target), where distributions remain similar, naive EM tends to be limited or even counterproductive. In these scenarios, energy‑based optimization can offer a solution by enhancing local observability. While EM struggles to detect how the probability landscape shifts around individual test samples, lowering a sample’s energy via SSM pulls not only that sample but also nearby points in the space into lower‑energy (higher‑likelihood) regions. This broader effect may help fill the observable gaps that EM alone often misses and mitigates misclassifications. Therefore, when the distribution overlaps between source and target domains are significant, incorporating minimization of the energy may provide an opportunity to alleviate the penalties incurred by EM substantially.
>
> &nbsp;
>
> This feedback helped us uncover that EM performs well under large shifts (e.g., ImageNet‑C) but underperforms when source and target domains are highly overlapped (e.g., text sentiment). This study allowed us to rethink entropy and energy again: by lowering energy, we can both improve adaptation performance and compensate for EM’s blind spots in overlap scenarios.
>
> &nbsp;
>
> > Minor comments
>
> Thank you for your careful reading and helpful suggestions. We will add a description for Fig. 1d in its caption (refer to the caption of Fig. 2 in the Appendix for an example), bold the highest DeYO values in the accuracy tables, and clearly distinguish entropy minimization from energy minimization throughout the manuscript to improve clarity.
>
> &nbsp;
>
> ---
> [1] Sanh et al., DistilBERT, a distilled version of BERT: smaller, faster, cheaper and lighter, NeurIPS Workshop, 2019

---

> > ### Comment · Reviewer_YKNW · 2025-08-04
> >
> > Thank you for your response and I appreciate the new experimental results that have been presented. My main concern was related to ReTTA's generalization to other Image datasets (similar to what reviewer-34Nx has pointed out) and has been addressed in the response to reviewer-34Nx. However, the results on text-based sentiment classification show that ReTTA (and also Tent TTA method), in their current form for image-based tasks, are not effective for text-based tasks (compared to un-adapted results). I would suggest to add these results to the paper (or at least to the Appendix if space does not permit) and provide a discussion about it in the limitation section (pointing to the results in the Appendix).
> >
> > I am keeping my score the same.

---

> > > ### Author Response · Authors · 2025-08-05
> > >
> > > Thank you for your careful and encouraging feedback. We truly appreciate your continued recognition of our work. Your suggestions are valuable for improving the final version of our manuscript. As recommended, we will include the sentiment classification results in the Appendix and discuss them in the limitation section.
> > >
> > > &nbsp;
> > >
> > > Sincerely,
> > >
> > > The Authors

---

### Official Review · Reviewer_GM3k · 2025-07-07

**Clarity:** 2
**Significance:** 2
**Originality:** 2
**Rating:** 4
**Confidence:** 4

**Summary:**

The paper revisits test-time adaptation (TTA) through the lens of energy–entropy duality, showing that classic entropy minimization can stall in high-energy regions and thus fails to maximise test-sample likelihood. To overcome this gap, the authors introduce ReTTA, which couples entropy minimisation with (i) a sampling-free Sliced Score Matching energy loss and (ii) a Targeted Class Convergence cross-entropy loss, balanced by a self-adjusting coefficient. Extensive experiments on ImageNet-C with both ResNet-50 and ViT-Base demonstrate the effectiveness of the proposed method.

**Questions:**

Please refer to the Weaknesses.

**Ethical Concerns:**

["NO or VERY MINOR ethics concerns only"]

**Final Justification:**

I appreciate the authors’ detailed responses. The authors have addressed most of concerns, such as more experiments, more clarification about the overconfidence. However, the memory overhead and inference latency remain relatively high.

That said, considering the overall contribution and the improvements made, I will increase my score.

**Quality:**

3

**Strengths And Weaknesses:**

Positive Points:
1. The energy–entropy duality analysis pinpoints why entropy minimisation alone is insufficient and directly motivates the SSM and TCC objectives.
2. ReTTA is tested across two architectures, two normalisation schemes and multiple corruption and label-shift scenarios, consistently surpassing state-of-the-art baselines.

Negative Points:
1. The Sliced Score Matching component introduces second-order Jacobian terms, yet the paper provides no comparison of runtime or memory consumption against standard entropy minimisation (EM) or energy-based TTA baselines such as TEA.
2. Entropy minimization may encounter the overconfindence issues[1]. The proposed Targeted Class Convergence loss adopts the current top-1 output as a pseudo-label, which could amplify this mis-calibration and accumulate errors under heavy domain shift; the manuscript neither analyses the phenomenon nor proposes safeguards.
3. The influence of the λ₁ weight on SSM, as well as the dynamics of the self-adjusting coefficient, are not ablated, leaving their robustness and default settings unclear.
4. While the method is evaluated on ViT-Base under severe ImageNet-C corruption, results for the same architecture in the mild setting are absent, limiting evidence of effectiveness across shift intensities.

[1] come: test-time adaption by conservatively minimizing entropy. ICLR 2025

---

> ### Author Rebuttal · Authors · 2025-07-31
>
> We appreciate your valuable time and effort in providing constructive feedback. We believe that clarifications in response to the concern raised enhance the quality of this manuscript and hope for more favorable consideration of our research.
>
> &nbsp;
>
> >W1. The Sliced Score Matching component introduces second-order Jacobian terms, yet the paper provides no comparison of runtime or memory consumption against standard entropy minimisation (EM) or energy-based TTA baselines such as TEA
>
> We appreciate the reviewer’s careful feedback. While Tent (EM-only) incurs minimal overhead, energy-based approaches generally involve higher memory and runtime demands. For example, TEA, which applied only an energy‑based contrastive divergence (CD) loss, requires approximately 8 GB of memory and exhibits significant runtime due to its generative procedures. In comparison, our combined EM + SSM implementation uses around 10 GB, showing that while the energy‑based method is inherently more costly than EM, our SSM adds only a modest overhead. Notably, our method runs about 30% faster than TEA, thanks to optimized Jacobian/Hessian computations that outperform TEA’s generative CD updates on batch data.
>
> | Method            | Peak GPU Memory (MB) | Runtime (s) |
> |-------------------|-----------------:|-----------------:|
> | **Tent (EM)**     |              2863 |            173 |
> | **TEA (Energy‑only CD loss)** |    8136  |            **3553** |
> | **Ours (EM + SSM)** |          **10439**  |            2480 |
>
> _Tested on: Intel(R) Xeon(R) Silver 4214 @ 2.20GHz CPU, a single NVIDIA TITAN RTX 24GB GPU_
>
> We emphasize that this work prioritizes algorithmic contributions as a foundation for future optimizations. As energy-based TTA continues to mature, we expect engineering advances—such as memory-efficient formulations, faster approximations, and improved system-level support—to follow naturally.
>
>
> &nbsp;
>
> >W2. The proposed Targeted Class Convergence loss adopts the current top-1 output as a pseudo-label, which could amplify this mis-calibration and accumulate errors under heavy domain shift; the manuscript neither analyses the phenomenon nor proposes safeguards
>
> We did not intend TCC to serve as a standalone approach. Rather, we were motivated that EM alone could not guess a test sample’s neighborhood or accurately estimate its likelihood, so we first applied SSM to pull samples (and their neighbors) into high‑likelihood regions. We could then use TCC to sharpen class probabilities on these refined representations. By integrating our losses into state-of-the-art EM workflows, we also inherited an entropy-based data filtering step, adapting only to low-entropy (and thus highly reliable) samples, to naturally guard against error amplification. In this design, SSM provided the core likelihood safeguard, data filtering prevented error accumulation, and TCC delivered the final discriminative push. As Table 3 shows, coupling EM with entropy‑based data filtering alone yielded dramatic improvements of 7% or 20% on corruptions like Glass or Frost, and on average delivered over a 1% gain versus the EM method. Also, the full ReTTA with SSM produced an average increase of over 2.5% relative to the EM method. It confirms that these components work in concert for robust, strong adaptation performance.
>
> &nbsp;
>
> >W3. The influence of the $\lambda_1$ weight on SSM, as well as the dynamics of the self-adjusting coefficient, are not ablated, leaving their robustness and default settings unclear
>
> We note that our self‑adjusting coefficient is exactly $\lambda_1$, which by design adapted automatically to any distribution shift—so a fixed‑value ablation was unnecessary. As shown in Figure 1, the bar plot tracked $\lambda_1$ across different corruption types and batch iterations, demonstrating how it dynamically scaled the SSM gradient without a manual option.
>
> &nbsp;
>
> >W4. While the method is evaluated on ViT-Base under severe ImageNet-C corruption, results for the same architecture in the mild setting are absent, limiting evidence of effectiveness across shift intensities.
>
> To ensure fairness, we used the DeYO source code repository to run the full suite of 15 severity‑5 corruption evaluations on ViT‑Base (LayerNorm) ourselves. The complete results appear below. Even under this mild setting, ReTTA led on average, achieving the highest accuracy on 10 of the 15 corruptions and second‑best on the remaining five. This confirmed that, across different model architectures, under severe distribution shifts, the simultaneous contributions of energy minimization and discriminability were crucial. Due to time constraints, we could not extend this evaluation to all five severity levels; instead, we performed additional tests on ResNet‑50‑BN at severities 1 and 3 (for Reviewer 34Nx), which mirrored these performance trends and further validated ReTTA’s robustness across varying shift intensities. We observed that at lower severities, all TTA methods delivered similar average performance (with ReTTA only marginally better), whereas as severity increased, jointly minimizing energy became essential to sustain high accuracy.
>
> | Method          | Gauss. | Shot  | Impul. | Defoc. | Glass | Motion | Zoom  | Snow  | Frost | Fog   | Brit. | Contr. | Elastic | Pixel | JPEG  | Avg  |
> |-----------------|-------:|------:|-------:|-------:|------:|-------:|------:|------:|------:|------:|------:|-------:|--------:|------:|------:|-----:|
> | No adapt        |   35.1 |  32.2 |   35.9 |   31.4 |  25.3 |   39.4 |  31.6 |  24.5 |  30.1 |  54.7 |  64.5 |   49.0 |    34.2 |  53.2 |  56.5 | 39.8 |
> | Tent            |   51.9 |  51.5 |   53.1 |   51.8 |  47.8 |   56.3 |  49.4 |  10.6 |  18.3 |  67.1 |  73.3 |   66.7 |    51.5 |  65.2 |  64.5 | 51.9 |
> | EATA            |   **55.9** |  **56.4** |   **57.1** |   53.3 |  53.5 |   58.2 |  **58.5** |  61.9 |  60.1 |  71.4 |  75.3 |   **68.5** |    62.4 |  69.5 |  66.6 | 61.9 |
> | SAR             |   51.8 |  51.7 |   52.9 |   50.8 |  48.9 |   55.6 |  49.1 |  12.8 |  51.0 |  65.8 |  73.1 |   66.1 |    51.9 |  64.0 |  63.3 | 53.9 |
> | DeYO            |   54.5 |  55.1 |   55.8 |   53.8 |  54.6 |   62.4 |  58.0 |  64.0 |  63.3 |  71.6 |  77.3 |   67.3 |    65.6 |  71.5 |  68.3 | 62.9 |
> | TEA             |    8.8 |  20.2 |   11.8 |    1.9 |   4.2 |   14.7 |   4.4 |   1.2 |   3.6 |   8.9 |  73.4 |   62.8 |     3.0 |  67.3 |  64.2 | 23.3 |
> | **ReTTA**       |   55.1 |  55.9 |   56.5 |   **56.4** |  **55.9** |   **62.5** |  57.9 |  **64.6** |  **63.5** |  **72.2** |  **77.4** |   67.4 |    **65.7** |  **71.6** |  **68.6** | **63.4** |
>
> We hope these clarifications have effectively addressed your concerns and provided a clearer understanding of our contributions.

---

> > ### Comment · Reviewer_GM3k · 2025-08-06
> > **Response to Rebuttal**
> >
> > I appreciate the authors’ detailed responses. The authors have addressed most of concerns. However,  the memory overhead and inference latency remain relatively high.
> >
> > That said, considering the overall contribution and the improvements made, I will increase my score.

---

> ### Author Response · Authors · 2025-08-06
>
> Thank you for your valuable and encouraging feedback. We truly appreciate your recognition of our improvements and overall contribution. Also, we are glad that our responses helped address most of the concerns.
>
> &nbsp;
>
> Sincerely,
>
> The Authors

---

### Note · Authors · 2025-08-14

We sincerely thank the Area Chair and the reviewers for their valuable and encouraging feedback, as well as their time and attention throughout the review process.

This work stems from the recognition that minimizing entropy alone in test-time adaptation may not sufficiently bridge the crucial likelihood gap between the trained and test distributions under covariate shifts. This realization helped us isolate the role of entropy and sparked the design of our new approach. Our method, ReTTA, rooted in enhanced entropy minimization, energy-based adaptation, and a self-balancing mechanism, sets our work apart and significantly contributes to the field. We are truly grateful that all reviewers recognized the contributions and improvements made.

We value the constructive suggestions for further strengthening the work. As committed in our post-rebuttal response, we will incorporate the requested improvements, including additional experimental results, analyses, and an expanded discussion in the limitation section. These updates will clarify the scope of our method, address remaining concerns, and certainly enhance the quality of the paper.

We deeply appreciate the opportunity to share our work with this esteemed community and look forward to engaging in meaningful discussions during the conference. Hopefully, this work will contribute to the continued commitment to meeting the high standards of NeurIPS.

&nbsp;

Best regards,

The Authors

---

### Decision · Program_Chairs · 2025-09-17

**Decision:**

Accept (spotlight)

**Comment:**

This work introduces ReTTA, a novel test-time adaptation method that goes beyond standard entropy minimization to address the likelihood gap under covariate shifts. By combining enhanced entropy minimization, energy-based adaptation, and a self-balancing mechanism, ReTTA achieves more reliable and effective adaptation, marking a distinct contribution to the field.

Initially, the reviewers raised several concerns regarding runtime efficiency, hyper-parameter sensitivity, effectiveness across diverse datasets, models, and tasks (including text and speech modalities), as well as broader scenarios such as lifelong adaptation and evaluations under mild shifts. They also requested clearer motivation and methodological details. During the rebuttal, the authors provided thorough discussions and additional empirical evidence, which substantially improved the paper’s quality and convincingly addressed concerns from all reviewers.

Overall, both the reviewers and the AC recognize the proposed method as simple yet effective, offering a new fully test-time adaptation framework with broad applicability. The motivation is theoretically grounded, and the empirical validation has been strengthened with extensive new results after the rebuttal. Accordingly, the AC recommends this work for a “spotlight” presentation.

Please ensure that the new rebuttal results and discussions are incorporated into the final version.

ps. The AC also encourages the authors to further explore and compare performance in other wild settings (e.g., batch size of 1 and mixed domain shifts). Such analyses would not only strengthen the justification of the method’s superiority but also offer valuable insights and comparative benchmarks for future work. The inclusion of these results and discussions in the final version is recommended.